# Single-cell transcriptomics of the naked mole-rat reveals unexpected features of mammalian immunity

**Hugo G. Hilton**[1][ø], **Nimrod D. Rubinstein**[1][ø], **Peter Janki**[1], **Andrea T. Ireland**[1],
**Nicholas Bernstein**[1], **Nicole L. Fong**[1], **Kevin M. Wright**[1], **Megan Smith**[1], **David Finkle**[1],
**Baby Martin-McNulty**[1], **Margaret Roy**[1], **Denise M. Imai**[2], **Vladimir Jojic**[1],
**Rochelle Buffenstein**[1]*

**1** Calico Life Sciences LLC, South San Francisco, California, United States of America, **2** Comparative
Pathology Laboratory, School of Veterinary Medicine, University of California, Davis, Davis, California, United
States of America

ø These authors contributed equally to this work.
* rbuffen@calicolabs.com

Cancer Institute, UNITED STATES

**Data Availability Statement:** Raw and processed
data can be downloaded from GEO under
accession number GSE132642. In addition,

## Abstract

The immune system comprises a complex network of specialized cells that protects against
infection, eliminates cancerous cells, and regulates tissue repair, thus serving a critical role
in homeostasis, health span, and life span. The subterranean-dwelling naked mole-rat (NM-
R; *Heterocephalus glaber*) exhibits prolonged life span relative to its body size, is unusually
cancer resistant, and manifests few physiological or molecular changes with advancing age.
We therefore hypothesized that the immune system of NM-Rs evolved unique features that
confer enhanced cancer immunosurveillance and prevent the age-associated decline in
homeostasis. Using single-cell RNA-sequencing (scRNA-seq) we mapped the immune sys-
tem of the NM-R and compared it to that of the short-lived, cancer-prone mouse. In contrast
to the mouse, we find that the NM-R immune system is characterized by a high myeloid-to-
lymphoid cell ratio that includes a novel, lipopolysaccharide (LPS)-responsive, granulocyte
cell subset. Surprisingly, we also find that NM-Rs lack canonical natural killer (NK) cells. Our
comparative genomics analyses support this finding, showing that the NM-R genome lacks
an expanded gene family that controls NK cell function in several other species. Further-
more, we reconstructed the evolutionary history that likely led to this genomic state. The
NM-R thus challenges our current understanding of mammalian immunity, favoring an atypi-
cal, myeloid-biased mode of innate immunosurveillance, which may contribute to its remark-
able health span.

## Introduction

The immune system plays a critical role in identifying and destroying pathogens and nascent
malignancies and is a key regulator in tissue homeostasis. Its age-associated decline in most
mammalian species is concomitant with an age-dependent increase in chronic diseases [1].

processed data are also provided as Supporting Information.

**Funding:** All authors of this manuscript, HGH, NDR, PJ, ATI, NB, NLF, KMW, MS, DF, BM-M, MR, DMI, VJ, and RB, as well as this work itself, have been entirely funded by Calico Life Sciences. The funders approved performing the study and its publication.

**Competing interests:** I have read the journal's policy and the authors of this manuscript have the following competing interests: NDR, ATI, NB, NLF, KMW, MS, BM-M, MR, VJ, RB, HGH, PJ, and DF were employees of Calico Life Sciences at the time that the study was undertaken.

**Abbreviations:** APC, antigen-presenting cell; Camp, cathelicidin; CD, cluster of differentiation; Crisp1, cysteine-rich secretory protein-1; EST, expressed sequence tag; FPKM, fragments per kilobase million; GO, gene ontology; GSEA, gene set enrichment analysis; HE, hematoxylin and eosin; IACUC, institutional animal care and use committee; ISH, in situ hybridization; KIR, killer cell immunoglobulin-like receptor; KNN, k-nearest neighbor; LPS, lipopolysaccharide; LRC, leukocyte receptor complex; Ltf, lactotransferrin; MHC-I, major histocompatibility complex class I; NK, natural killer; NKC, natural killer cell receptor complex; NF-κB, nuclear factor kappa-light-chain-enhancer of activated B cells; NM-R, naked mole-rat; Olfm4, olfactomedin-4; PAS, periodic acid–Schiff; PCA, principal components analysis; RPS, reverse position-specific; scRNA-seq, single-cell RNA-sequencing; SNN, shared nearest neighbor; TLR, toll-like receptor; TPM, transcripts per million; UMAP, uniform manifold approximation and projection; UMI, unique molecular identifier.

Unlike most mammals, the naked mole-rat (NM-R), a mouse-sized rodent, is cancer-resistant [2], extremely long-lived relative to its body size [3], and shows no age-associated exponential increase in risk of dying [4], suggesting it maintains cellular and systemic homeostasis well into its third decade of life [5,6]. An unexplored possibility is that the NM-R has evolved features of systemic immunity that contribute to these remarkable phenotypes.

Investigating the immune systems of non-model organisms, such as the NM-R, has traditionally been constrained by the lack of species-specific reagents. To circumvent these limitations, we used single-cell RNA-sequencing (scRNA-seq) to obtain an unbiased molecular profile of the immune systems of the NM-R and the well-characterized mouse, allowing us to compare their immune cell repertoires. We find that the cellular composition of the immune system of the NM-R is markedly different from that of the mouse. These differences include an inverted myeloid-to-lymphoid cell ratio, a novel lipopolysaccharide (LPS)-responsive granulocyte cell subset, and, most surprisingly, the absence of natural killer (NK) cells. The latter feature is supported by comparative genomics analyses, showing a lack of diversity in the gene families controlling NK cell function in other mammalian species.

## Results

### The immune systems of the NM-R and mouse have strikingly different cellular compositions

In order to obtain cell repertoires of the NM-R and mouse immune systems, spleens were harvested from young healthy C57BL/6 mice (2–3 months old; *n* = 4) and NM-Rs (approximately 2 years old; *n* = 4) and a 10× Genomics [7] droplet-based scRNA-seq library was prepared from each organ (S1A–S1D and S1H–S1K Fig). Gene-by-cell expression data were clustered separately for each species using an iterative approach (Materials and methods; S1 and S2 Tables). Subsequently, sets of marker genes were derived for each cluster (S3 and S4 Tables), thereby allowing their annotation and revealing the major immune cell subsets present in each species (Materials and methods; Fig 1A–1D; S1E, S1G, S1L and S1N Fig).

Consistent with previous reports [8,9], the splenic immune cell population of mice is dominated by lymphoid lineage cells (approximately 90% of the cells; Fig 1A and 1C). B cells (e.g., marked by *Cd19* and *Cd79a*) comprise 59% of the total splenic immune cell population; memory B cells (additionally marked by *Try5*) make up 2%, T cells (e.g., marked by the four cluster of differentiation [CD] 3 subunit genes: *Cd3g*, *Cd3e*, *Cd3g*, and *Cd247*) 28%, and NK cells (e.g., marked by *Ncr1*) 3%. Myeloid lineage cells account for the remaining 8%, breaking down to 7% antigen-presenting cells (APCs; e.g., marked by *Cd68*, *Ccl6*, and *C1qa*), which resolve to macrophage and dendritic cell subsets at the converged clusters (S1E and S1F Fig), and 1% neutrophils (e.g., marked by *Cxcr2*).

By contrast, lymphoid lineage cells comprise only 44% of the splenic immune cell population in NM-Rs (Fig 1B and 1D). Specifically, T cells (e.g., marked by the four CD3 subunit genes: *Cd3d*, *Cd3e*, *Cd3g*, and *Cd247*) account for 30% and B cells (e.g., marked by *Cd19* and *Cd79a*) 14%. The remaining 56% of the splenic immune cell population are myeloid lineage, with 10% macrophages (e.g., marked by *Cd68*, *C1qa*, and *C1qb*), 6% red-pulp macrophages (additionally marked by *Vcam-1* [10]), and 37% neutrophils (e.g., marked by *Cxcr2*). Two separate clusters of neutrophils were identified; the larger cluster accounted for 36% of the cells and the smaller cluster, defined by not only the neutrophil expression profile but also by high expression of the antimicrobial lactotransferrin (*Ltf*), olfactomedin-4 (*Olfm4*), cathelicidin (*Camp*), and cysteine-rich secretory protein-1 (*Crisp1*), accounted for 1% of the splenic immune cells. We have termed these cells *Ltf*-high neutrophils. Dendritic cells (e.g., marked by *Cd74*) contribute to 2% of the splenic immune cells. These cells also showed high expression of

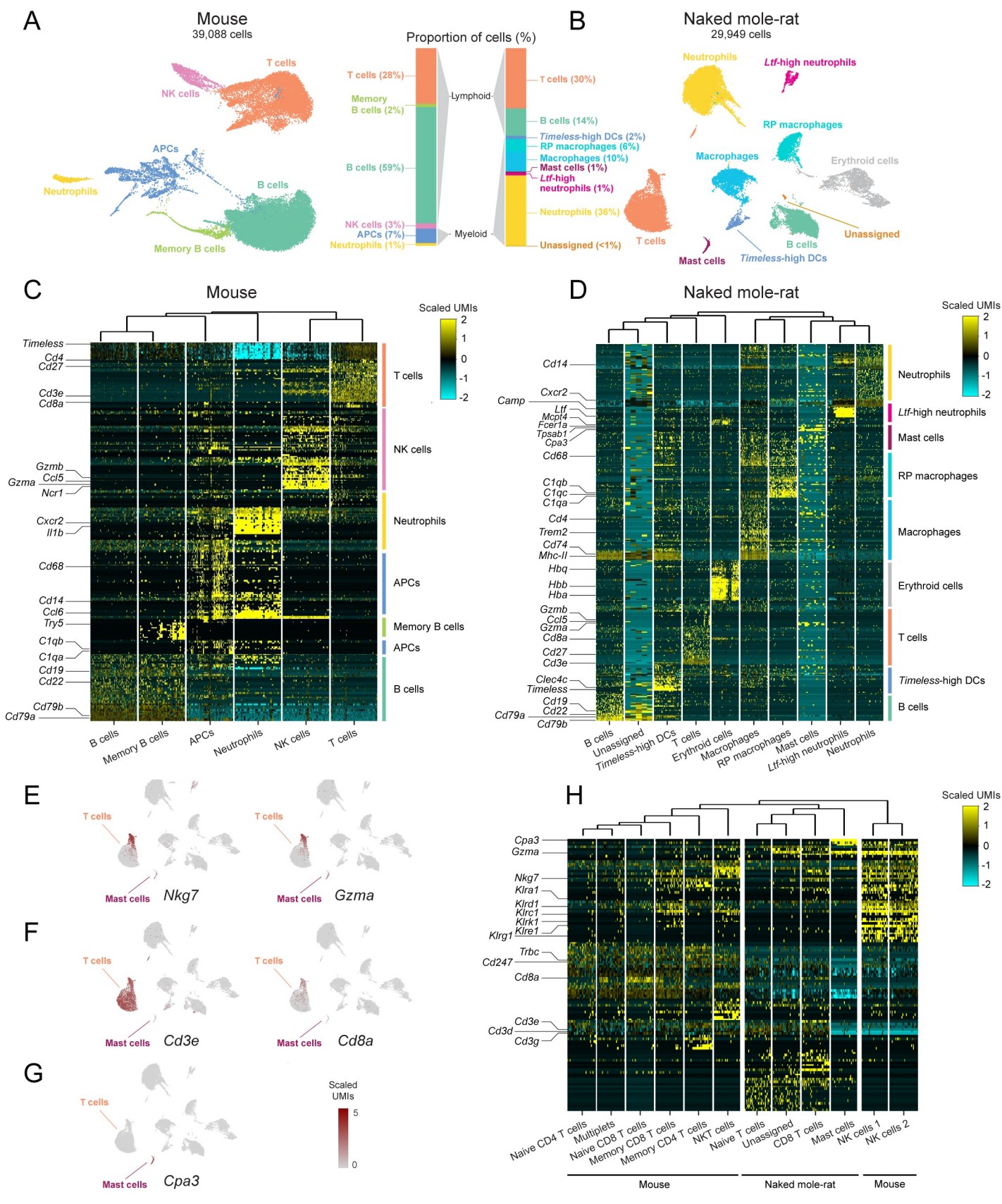

**Fig 1. scRNA-seq of mouse and NM-R spleens reveals major differences in immune cell populations.** UMAP projections of four mouse (**A**) and four NM-R (**B**) spleens. Each point is a cell color-coded by its cluster assignment and annotated cell type (Materials and methods). The proportions (%) of cells assigned to each cell type in each species (excluding NM-R clusters of erythroid cells) are shown in the stacked bar charts. Gene-by-cell expression-level heatmaps in the mouse (**C**) and NM-R (**D**) spleens. Selected marker genes are listed to the left and cells are faceted by their cluster assignment. UMAP projections of the NM-R spleens color-coded by the expression levels of *Nkg7* and *Gzma* (**E**), *Cd3e* and *Cd8a* (**F**), and *Cpa3* (**G**). Gene-by-cell expression-level heatmap of the mouse spleen T cells and NK cell clusters and NM-R spleen T cells and mast cell clusters (**H**). APC, antigen-presenting cell; DC, dendritic cell; NK, natural killer; NKT, natural killer T; NM-R, naked mole-rat; RP, red pulp; scRNA-seq, single-cell RNA-sequencing; UMAP, uniform manifold approximation and projection; UMI, unique molecular identifier.

*Timeless*, a gene associated with the regulation of circadian rhythm and protection of telomeres from DNA damage [11–13].

To confirm the interspecies differences we observed in our scRNA-seq data, we used comparative histomorphology and in situ hybridization (ISH; Materials and methods). This additionally revealed major interspecies differences in the distribution of the red and white pulp (S2A Fig), with significantly higher expression of *Cd19* (B cell marker) and lower expression of *Cd14* (myeloid lineage marker) in the mouse than in the NM-R (S2B–S2D Fig; S5 Table). Moreover, while T cells were confined to the white pulp in mice, they were distributed more widely in NM-R spleens.

## The immune system of the NM-R lacks canonical NK cells

Although lymphoid lineage cells were clearly detected in the NM-R spleen (Fig 1B and 1D; S1L, S1M and S2 Figs), surprisingly, a cluster corresponding to NK cells was not identified. In mice, NK cells are marked by high expression of several genes, including *Ncr1*, *Gzma*, *Ccl5*, and *Nkg7* (Fig 1C and 1H; S3 Table). In NM-Rs, expression of *Ncr1* was detected neither in the scRNA-seq data nor in whole spleen RNA-sequencing data. By contrast, we did find high expression of *Gzma*, *Ccl5*, and *Nkg7* in a cell subset we identified as part of the T-cell cluster (Fig 1D and 1H; S1M and S3A Figs). Similar to all other cell subsets within this T-cell cluster, this cell subset showed high expression of all four CD3 subunit genes: *Cd3d*, *Cd3e*, *Cd3g*, and *Cd247* and additionally is the only T-cell subset that showed high expression of *Cd8a*, consistent with its identity as CD8 T cells rather than NK cells (Fig 1D and 1H; S3B Fig; S6 Table). Mouse NK cells showed negligible expression of these T-cell markers [14] (Fig 1B and 1H; S1F Fig). Similarly, another NM-R cluster also showed high expression of *Gzma* and *Nkg7*, yet additionally showed high expression of *Cpa3*, *Tpsab1*, *Cma1*, and *Fcer1a*, in keeping with its identity as mast cells [15] rather than NK cells (Fig 1C and 1H; S1M and S3C Figs).

To exclude the possibility that the absence of NK cells in NM-Rs is specific to the spleen, we also performed scRNA-seq of circulating immune cells from an additional four C57BL/6 mice and four NM-Rs (Materials and methods; S4A–S4D Fig; S7–S10 Tables). A mouse NK cell cluster was readily detected, yet, again, NK cells were not detected in the NM-R (S4E–S4H Fig). Finally, we tested the possibility that the observed lack of NK cells in our NM-R scRNA-seq data is a result of poor annotation of their marker genes in the NM-R genome. Simulating this situation in the mouse (Materials and methods), NK cells were still detected in a distinct cluster (S5 Fig), suggesting that the absence of an NM-R NK cell cluster is not an artifact of genome annotation quality.

## The NM-R genome lacks expanded gene families that control NK cell function

We sought an evolutionary genomic explanation for the absence of NM-R NK cells and therefore examined the genes whose products control NK cell function. These include genes encoding NK cell receptors and their major histocompatibility complex class I (MHC-I) ligands [16]. In mammals, NK cell receptors are encoded in two distinct gene complexes: the leukocyte

receptor complex (*LRC*) and the natural killer cell receptor complex (*NKC*). The *LRC* predominantly contains genes encoding immunoglobulin superfamily receptors, including killer cell immunoglobulin-like receptors (*KIRs*) that have dramatically expanded in Old World monkeys and apes [17]; the *NKC* contains gene families encoding killer cell lectin-like receptors such as *Cd94* (*Klrd*), *Nkg2* (*Klrc*), and *Ly49* (*Klra*), where the latter has dramatically expanded in the murioids [18]. We compared the NM-R *LRC*, *NKC*, and *MHC-I* genes to those of the human and mouse, in which they are well characterized. Consistent with its phylogeny, the NM-R *LRC*, like that of the mouse, does not harbor an expanded *KIR* family (Fig 2A; S11 Table). However, in stark contrast to the mouse, the NM-R *NKC* does not have an expanded *Ly49* gene family, harboring only a single protein-coding *Ly49* gene (*LOC101714034*; a mouse *Klra1* ortholog). This gene, which is predicted to encode a classical inhibitory MHC-I receptor (S6A Fig), is expressed at low levels in NM-R CD8 T cells, whereas its mouse ortholog is highly expressed in mouse CD8 T cells, as well as in NK cells (S6B and S6C Fig, respectively). Moreover, unlike the diversity of *MHC-I* genes in the human and mouse genomes (with both 6 and 22 protein-coding genes, as well as 7 and 14 pseudogenes, respectively), the NM-R genome has only three *MHC-I* genes (two protein-coding genes and one pseudogene) (Fig 2C). These data suggest that NM-Rs lack the control mechanisms of canonical NK cells, supporting our scRNA-seq findings that NM-Rs lack NK cells.

In order to understand how widespread this lack of species-level diversity in NK cell receptor and *MHC-I* genes is, we expanded our analysis to include 45 additional mammalian genomes, among them several members of the same suborder of rodents to which the NM-R belongs (Hystricomorpha). Our analysis additionally included the detection of old and hence unannotated NK cell receptor and *MHC-I* pseudogenes in order to reconstruct their evolutionary histories (Materials and methods; S7 Fig, S12 and S13 Tables). This revealed that the *Ly49* gene family expanded from the single ancestral gene to three members prior to the Lagormorpha-Rodentia split (Fig 2D). Subsequently, the *Ly49* family went through several expansions and contractions along the muroid lineage, leading to the highly diversified *Ly49* gene family in the mouse and rat and the large array of pseudogenes in all other muroid genomes. This finding is largely mirrored by their *MHC-I* patterns (Fig 2D), consistent with their reported coevolution [16]. Interestingly, among the examined hystricomorphs, only the guinea pig retained all three ancestral *Ly49* genes, and in addition to its six protein-coding *MHC-I* genes, it was found to harbor 19 *MHC-I* pseudogenes (Fig 2D). By contrast, the NM-R as well as the other two hystricomorphs for which genome data are available lost either two or all three of their *Ly49* genes. Accordingly, their genomes were also found to harbor dramatically lower numbers of both protein-coding and pseudo *MHC-I* genes (Fig 2D), suggesting that within these hystricomorphs, only the guinea pig has retained the control mechanisms of canonical NK cells. Intriguingly, the guinea pig is known to have a population of atypical, histologically distinct mononuclear cells (Foa-Kurloff cells) with reported NK-like activity [19,20]. Although we readily detected these cells in a guinea pig peripheral blood smear (S6D Fig), we did not detect morphologically similar cells in NM-R peripheral blood smears (S6E–S6I Fig), further supporting the premise that NM-Rs lack NK cells. We cannot rule out, however, that other cell types may have NK-like activity in the NM-R.

## The immune system of the NM-R contains an LPS-responsive granulocyte cell subset

The abundance of myeloid lineage cells in the NM-R that strongly express the toll-like receptor (TLR) 4 co-receptor *Cd14* (which together recognize bacterial LPS and other pathogen-associated molecules [21]), as well as their striking absence of NK cells raises the question whether

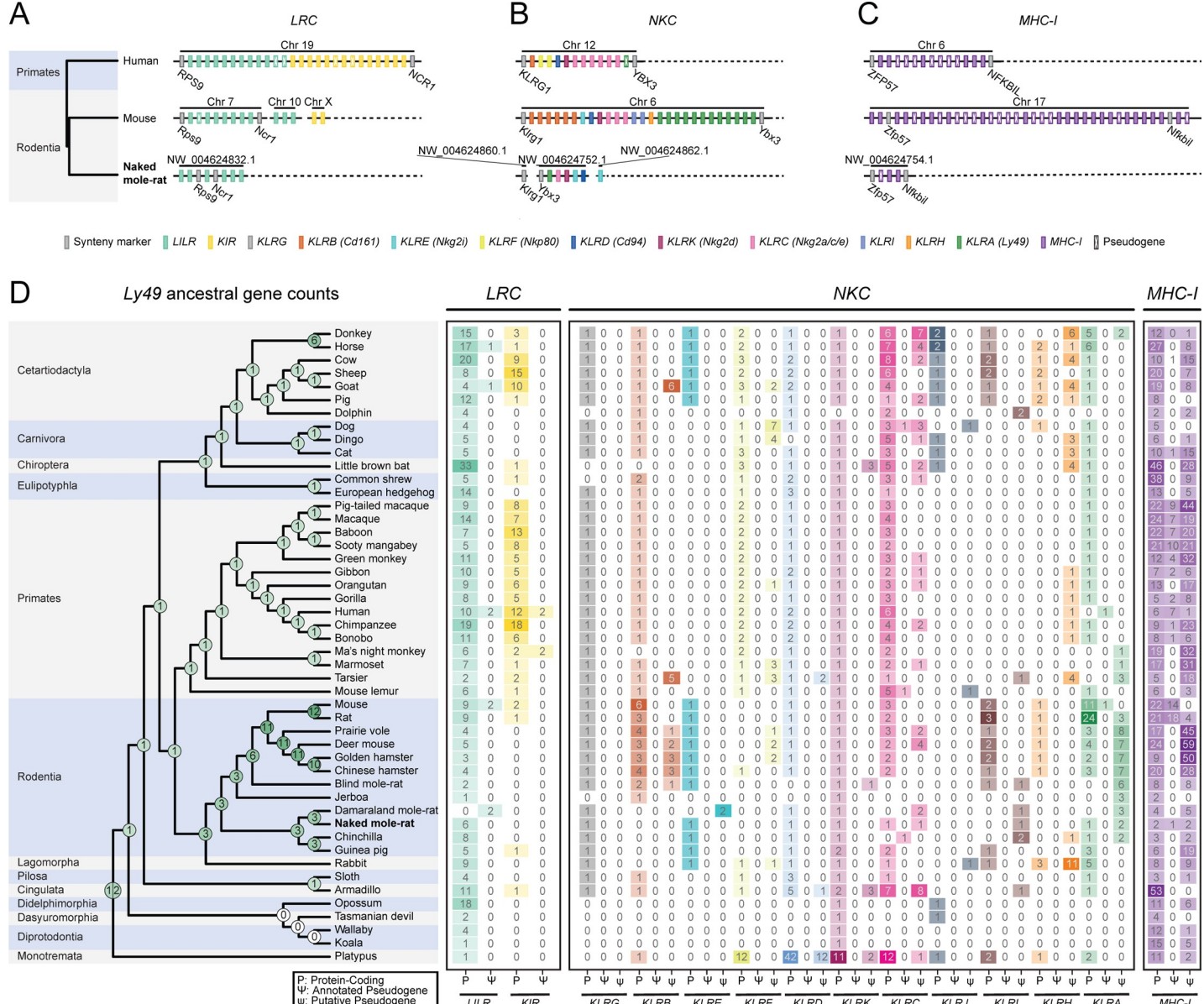

**Fig 2. The NM-R genome lacks expanded NK cell receptor and MHC-I gene families.** Schematic diagram of the linear arrangement of genes, in 5′-3′ genomic orientation relative to the human genome, in the *LRC* (**A**), *NKC* (**B**), and *MHC-I* loci (**C**) of the human, mouse, and NM-R genomes. Boxes represent the genes encoding NK cell receptors and their ligands—MHC-I, which are known to control NK cell function, color-coded by family. Synteny markers (gray boxes) are genes that flank the human *LRC*, *NKC*, and *MHC-I* loci; (**D**) phyletic pattern showing the numbers of protein-coding genes (P) and annotated pseudogenes (Ψ) (in the *LRC*, *NKC*, and *MHC-I* loci), as well as putative pseudogenes (ψ) (only in the *NKC* and *MHC-I* loci) across 48 mammalian genomes (Materials and methods). Overlaid on the phylogeny are the maximum parsimony–reconstructed ancestral numbers of the *Ly49* (*KLRA*) gene family. Color shades in both the phyletic pattern and ancestral *Ly49* counts correspond to the numbers of genes. Ancestral *Ly49* counts are represented as pie charts in which the slice sizes correspond to the probability of each count. In all phylogenies, order names are indicated and faceted by the gray and blue background colors. Chr, chromosome; *LRC*, leukocyte receptor complex; *MHC-I*, major histocompatibility complex class I; NK, natural killer; *NKC*, natural killer cell receptor complex; NM-R, naked mole-rat.

the NM-R immune system has evolved unique innate responses to bacterial infections. To address this question, we profiled gene expression from spleens and circulating immune cells in mice and NM-Rs following in vivo LPS administration, mimicking an acute bacterial infection.

Significant changes in gene expression following LPS challenge were evident in whole spleen RNA-sequencing (S8A–S8D Fig; S14 and S15 Tables). Gene set enrichment analyses (GSEAs) revealed that both species activated similar pathways, including the canonical nuclear factor kappa-light-chain-enhancer of activated B cells (NF-κB) inflammatory pathway (S8E and S8F Fig; S16 and S17 Tables). Spleen scRNA-seq revealed that cells from both saline control and LPS-challenged animals of both species clustered into similar cell types to those observed in untreated animals (Fig 3A and 3B; S18–S25 Tables). A differential gene expression analysis between LPS-challenged and saline control animals indicated changes in the proportion of cells expressing a certain gene (expression-induction) as the predominant driver of the transcriptional responses (S9A–S9G Fig; S26–S28 Tables).

The most striking species difference we observed following the LPS challenge was the change in the proportion of immune cells assigned to each NM-R cluster (Fig 3C and 3D). Most notable were the dramatic increase of *Ltf*-high neutrophils (from approximately 5% to >25%; adjusted $p < 10^{-22}$) and decrease in neutrophils (from 26% to 18%; adjusted $p < 10^{-22}$). The increase of *Ltf*-high neutrophils in LPS-challenged NM-Rs was accompanied by a convergence between the neutrophil and *Ltf*-high neutrophil clusters in the uniform manifold approximation and projection (UMAP) [22] embedding space (Fig 3B). In addition, unlike in the saline control, the trajectories of NM-R LPS-challenged neutrophils trended towards *Ltf*-high neutrophils (Fig 3B, 3E, and 3F). GSEA of genes differentially expressed between the saline control *Ltf*-high neutrophils and neutrophils indicated mainly down-regulation (adjusted $p < 0.1$) of chemotaxis pathways in *Ltf*-high neutrophils (or up-regulation in neutrophils) (Fig 3G; S29 Table). A similar analysis in the LPS-challenged NM-Rs showed up-regulation of humoral immune responses mediated by antimicrobial peptides and down-regulation of viral replication processes, host–pathogen interactions, and negative regulation of immune cell proliferation in *Ltf*-high neutrophils (both adjusted $p < 0.1$) (Fig 3H; S30 Table). Qualitatively similar results were observed in NM-R circulating immune cells (S10 Fig; S31–S42 Tables). These results strongly suggest that NM-Rs have two transcriptionally distinct but functionally integrated neutrophil populations that likely play an important role in innate antimicrobial immunity in this species.

## Discussion

Our investigation of the NM-R and mouse immune systems reveal striking differences in their cell repertoires. We find that the immune system of the NM-R is heavily reliant on innate myeloid lineage cells, lacks NK cells, and contains a previously undescribed, transcriptionally distinctive cell subset that is highly responsive to a bacterial-mimicking challenge.

To the best of our knowledge, our study is the first to report a mammalian species that lacks a clearly defined population of NK cells that form the first line of defense against viral infections and play an important role in cancer immunosurveillance [23]. The lack of NM-R NK cells is supported by a dearth of genes that regulate NK cell function (encoding NK cell receptors and their MHC-I ligands), a feature common to both the human and mouse genomes, in which NK cells have been well characterized. Notwithstanding, allelic and structural polymorphisms are common features of NK cell receptor and *MHC-I* genes [16], and because our study does not include population-level genomic data, their true genetic (and functional) diversity in the NM-R may be underestimated. However, population-level diversity typically adds to the species-level diversity of these genes (e.g., *KIR* in humans [16] and *Ly49* in mouse [18]), and a similar lack of gene diversity has been previously documented in several marine carnivores (seals and sea lions) at both the species and population levels [24]. In addition, our evolutionary analysis of NK cell receptor genes provides strong support to a scenario in which

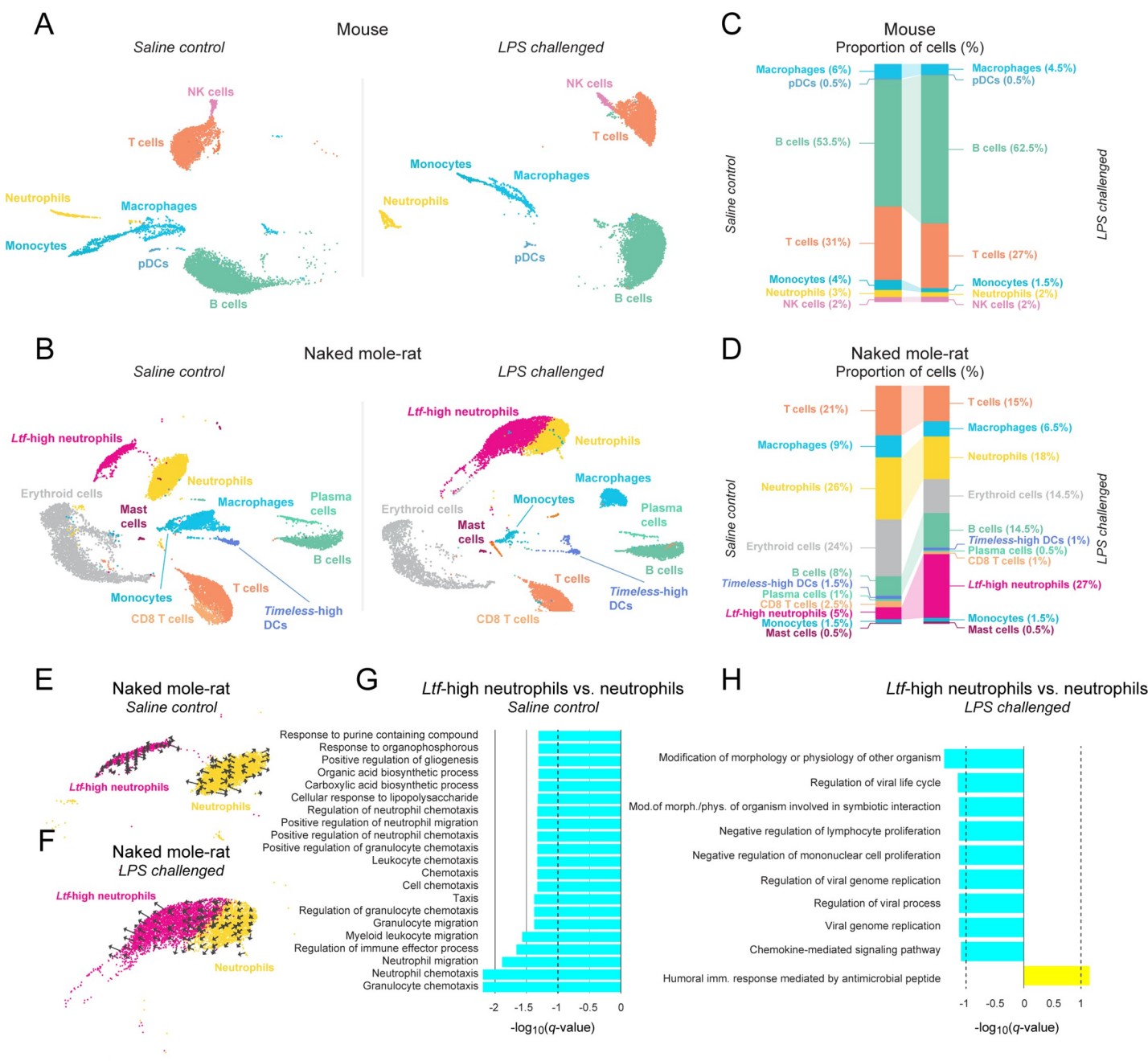

**Fig 3. Spleen scRNA-seq reveals that NM-Rs have an LPS-responsive cell subset not found in mice.** UMAP projections showing cell clusters from two saline control (left panel) and two LPS-challenged (right panel) mouse (**A**) and NM-R (**B**) spleens. Each point is a cell color-coded by its cluster assignment and annotated cell type (Materials and methods). Stacked bar charts show the proportions (%) of cells assigned to each cell type in the saline control (left) and LPS-challenged (right) mouse (**C**) and NM-R (**D**) spleens. UMAP projections showing cell clusters representing *Ltf*-high neutrophils (pink) and neutrophils (yellow) from saline control (**E**) and LPS-challenged (**F**) NM-R spleens. Overlaid on each cluster are arrows that represent the cell trajectory (see Materials and methods). Bar charts show GSEAs comparing *Ltf*-high neutrophils to neutrophils from saline control (see S29 Table for underlying data) (**G**) and LPS-challenged (see S30 Table for underlying data) (**H**) NM-R spleens. X-axes are the GSEA log$_{10}$-adjusted *p*-values (*q*-values), signed and color-coded by the direction of the effect (up- and down-regulation in *Ltf*-high neutrophils versus neutrophils: yellow and cyan, respectively). Black vertical dashed lines mark adjusted *p* = 0.1. DC, dendritic cell; GSEA, gene set enrichment analysis; LPS, lipopolysaccharide; NK, natural killer; NM-R, naked mole-rat; pDC, plasmacytoid dendritic cell; scRNA-seq, single-cell RNA-sequencing; UMAP, uniform manifold approximation and projection.

a common hystricomorph ancestor had NK-like cells, which, along with *Ly49* pseudogenization, were subsequently lost in NM-Rs and other hystricomorph rodents, possibly in response to a relaxation in pathogenic selective pressures. The low *MHC-I* gene diversity in these hystricomorphs supports this scenario, as high numbers of protein-coding and pseudo *MHC-I* genes are reflective of high rates of gene birth and death in response to pathogens [25].

The lack of NK cells and accompanying low diversity in the genes encoding their control mechanisms, together with the predominance of the myeloid lineage and its novel LPS-responsive, antimicrobial gene–expressing neutrophil-like subset, indicate that NM-Rs strongly rely on myeloid-based innate immunity. Taken together, these suggest that the NM-R immune system has evolved under stronger antibacterial than antiviral immune selective pressures. Additional support for this conclusion are two reports that NM-Rs are highly susceptible to viral infection [26,27]. Given their ecological niche and atypical mammalian behavior, it is possible that these xenophobic, eusocial, and strictly subterranean rodents naturally experience weaker viral selective pressures than other species such as bats, which live in open, typically crowded colonies, where intricate antiviral defenses are essential [28].

Finally, the manner by which its unusual immune features impact upon better maintenance of tissue homeostasis, prolonged health span, and especially cancer resistance in the NM-R and possibly other species with similar immune features remains to be elucidated.

## Materials and methods

### Ethics statement

All animal use and experiments were approved by the Buck Institute institutional animal care and use committee (IACUC) protocol number A10173.

### Animals

In this study, we used comparative histomorphology, comparative genomics, fluorescent ISH, and whole-tissue as well as single-cell gene expression to study the immune system of the NM-R in comparison with that of the mouse and other mammalian species. The animals used for RNA-seq in this study comprised 12 young C57BL/6 mice (8 to 10 weeks old; six males, six females, all virgins) and 12 young NM-Rs (23 to 25 months old; six males, six females, all nonbreeders). For blood smears, two young female guinea pigs (approximately 2 years old) and an additional five young nonbreeding NM-Rs (2 to 15 months old; three males, two females) were used. The NM-Rs in this study came from 10 different captive colonies, while the mice were purchased from the Jackson Laboratories (Bar Harbor, ME) and maintained in the vivarium for at least 2 weeks prior to use. The age range we selected was targeted to use young, healthy individuals that were physiologically age-matched between the species (approximately 5%–10% of observed maximum life span). Both species were maintained on a 12-hour dark-light cycle and provided food ad libitum. Mice were provided water ad libitum. In accordance with standard colony management, NM-Rs did not receive supplemented water, as the water content of their diet is sufficient for maintaining appropriate hydration.

### Organ collection and processing

We investigated the immune cell repertoire of eight NM-Rs (four males, four females) and eight C57BL/6 mice (four males, four females) using scRNA-seq of the spleen and circulating immune cells. Organ collection was performed on all animals between 8 AM and 10 AM. Animals were anesthetized using isoflurane. Blood was drawn via cardiac puncture and immediately transferred into EDTA-containing tubes, mixed by inversion, and placed on ice.

1. Blood: One milliliter of EDTA-treated blood was resuspended in 20 mL ACK lysis buffer (Gibco A1049201, Waltham, MA) for 5 minutes at 21˚C. Cells were pelleted (500$g$, 10 minutes, 4˚C), resuspended in PBS with 5% FBS, and washed twice. Following resuspension, cells were passed through a 40-μm filter and placed on ice prior to determination of viability and density. Visual inspection of trypan blue (Gibco 15250061, Waltham, MA) stained cells loaded on a hemocytometer slide was used to determine the density and viability of cells in the suspension.

2. Spleen: Immediately following cardiac exsanguination, the spleen was removed by dissection, transferred to a sterile petri dish containing PBS with 5% FBS, and minced using razors. Following trituration with a 5-mL serological pipette, the spleen fragments were ground through 100-μm and 40-μm cell strainers (Falcon 352360 and 352340, Corning, NY) with a syringe plunger. Cells were pelleted (500$g$, 10 minutes, 4˚C), resuspended in 20 mL of ACK lysis buffer (Gibco A1049201, Waltham, MA) for 5 minutes at 21˚C, and washed twice with PBS with 5% FBS. Following resuspension, cells were passed through a 40-μm filter and placed on ice prior to determination of viability and density. Visual inspection of trypan blue (Gibco 15250061, Waltham, MA) stained cells loaded on a hemocytometer slide was used to determine the density and viability of cells in the suspension.

## LPS challenge

In order to study the transcriptional response of mice and NM-Rs to stimulation of the innate immune system, we performed transcriptional profiling 4 hours following an LPS challenge. Two animals from each species (one male, one female) were randomly assigned into either LPS challenge or saline control groups (total = four mice, four NM-Rs). Saline control animals received intraperitoneal injection of 200 μL sterile saline. LPS-challenged animals received intraperitoneal injection of 1 mg/kg LPS (LPS-EB VacciGrade, *Escherichia coli* 0111:B4, Invivogen, San Diego, CA), prepared in sterile saline to a final volume of 200 μL. The body temperature of all animals was monitored by rectal probe every 30 minutes for 2 hours following injection. Collection and processing of blood and spleen were performed 4 hours following initial injection and as described above, with the exception that approximately one fifth of each spleen was immediately flash-frozen in liquid nitrogen (for whole-tissue RNA extraction and sequencing).

## scRNA-seq and data analysis

Single cells were captured in droplet emulsion using the Chromium Controller (10× Genomics, Pleasanton, CA), and scRNA-seq libraries were constructed according to the 10× Genomics protocol using the Chromium Single-Cell 3′ Gel Bead and Library V2 Kit (10× Genomics, Pleasanton, CA). In brief, cell suspensions were diluted in PBS with 5% FBS to a final concentration of $1 \times 10^6$ cells/mL (1,000 cells per μL). Cells were loaded in each channel with a target output of 3,000 cells per sample. All reactions were performed in a C1000 Touch Thermal Cycler (Bio-Rad Laboratories, Hercules, CA) with a 96 Deep Well Reaction Module. Twelve cycles were used for cDNA amplification and sample index PCR. Amplified cDNA and final libraries were evaluated using a Bioanalyzer 2100 (Agilent Technologies, Santa Clara, CA) with a high sensitivity chip. Samples were sequenced on an HiSeq 4000 instrument (Illumina, San Diego, CA).

scRNA-seq fastq files were demultiplexed to their respective barcodes using the 10× Genomics Cell Ranger mkfastq utility. Unique molecular identifier (UMI) counts were generated for each barcode using the Cell Ranger count utility. The mm10 reference genome and the

mouse Gencode version M12 annotation were used for mapping the mouse reads [29]. The HetGla2.0 reference genome along with the RefSeq GCF_000247695.1 annotation, to which we manually added the annotation of the *Ncr1* gene from Ensembl's *H. glaber* female genome annotation (which is the only NM-R genome annotation that covers this gene), were used for mapping the NM-R reads. For each sample, barcodes that were not likely to represent captured cells were filtered out by detecting the first local minimum above 2 in a distribution of $\log_{10}(\#UMIs)$. Similarly, for each sample, genes that were too sparsely captured across barcodes were filtered out by detecting the first local minimum above 3 in a distribution of $\log_{10}(\#barcodes)$. Finally, barcodes capturing more than a single cell (multiplets) were sought as local modes in the distributions of $\log_{10}(\#genes)$ and $\log_{10}(\#UMIs)$, whose x-axis maxima are more than 1.5 times higher than the x-axis location of the global maximum of the respective distribution and include less than 5% of the total number of barcodes. In other words, barcodes that, based on the number of genes or UMIs they captured, appeared inflated with respect to all other barcodes, were regarded as multiplets and thus filtered out (summary statistics shown in S43 Table).

To identify cell clusters, the samples from each species were concatenated, UMIs were then scaled to the read depth of their respective barcodes, and genes with high expression dispersion were obtained using Seurat [30] (S43 Table). Principal components analysis (PCA) was then performed on these gene-by-cell–scaled UMI matrices using the rsvd [31] R [32] package to reduce their genes dimension, retaining the 50 PCs explaining the highest amount of variation. We then used Seurat's methodology to build a shared nearest neighbor (SNN) graph of these cell-embedding data, first generating a k-nearest neighbor (KNN) graph using k = min(750, #cells-1) and a Jaccard distance cutoff of 1/15. The SNN graph was then used as input to the Louvain algorithm, implemented in the ModularityOptimizer software [33]. Because this implementation uses a resolution parameter that strongly affects the number of clusters, we searched the 0.05–1.225 range of this parameter, using the mean unifiability isolability clustering metric as our maximization parameter. This process was initially done on all cells in our data and subsequently repeated for each cluster individually, in an iterative manner in which convergence was defined as not being able to break down a cluster into subclusters (see S1, S2, S7, S8, S18–S21 and S31–S34 Tables for the cell-cluster assignments for each species-tissue-condition dataset).

In order to obtain gene markers for each cluster, we applied a differential expression test implemented in Seurat (using a likelihood ratio test between a model that assumes that a gene's expression values of two compared clusters were sampled from two distributions, versus a null model, which assumes they were sampled from a single distribution). The clusters that were used for this procedure were the ones obtained following convergence of our iterative clustering approach. A marker gene of a given converged cluster was thus defined as a gene that was found to be significantly overexpressed (multiple-hypotheses–adjusted $p < 0.05$) in that converged cluster compared to all other converged clusters (see S3, S4, S9, S10, S22–S25 and S35–S38 Tables for the converged cluster marker genes for each species-tissue-condition dataset). In all genes-by-cells heatmap figures, the total number of cells was reduced to a maximum of 2,500 (in proportions with respect to the number of cells per cluster) by randomly down-sampling cells in each cluster.

Cell types were assigned to the transcriptionally defined clusters in our data by intersecting the sets of marker genes we obtained for each cluster with immune cell expression data in transcripts per million (TPM) units, downloaded from the ImmGen portal [34], obtained by RNA-seq of sorted mouse immune cell types (S44 Table). Based on the ImmGen expression data, we subsequently only used the marker genes that were found to be strongly and specifically expressed in a specific ImmGen cell type. As a result, visualizing the ImmGen TPM expression

levels of the retained marker genes for each cluster clearly identified the cell types corresponding with the first-iteration clusters.

In both the spleen and circulating immune cell NM-R scRNA-seq datasets, our clustering approach isolated one or more clusters that showed high expression levels of the beta and alpha adult hemoglobin genes. However, slightly lower expression levels of these two genes were also observed in all other clusters (S11A, S11D and S12A, S12D Figs in spleen and circulating immune cells, respectively), which probably represents RNA released from lysed red blood cells, yet captured and sequenced by essentially all barcodes. Notwithstanding, the same clusters that showed high expression levels of the beta and alpha hemoglobin genes also showed isolated lower expression levels of the embryonic and fetal hemoglobin genes (epsilon, theta, and zeta, yet not of gamma) (S11B, S11C and S12B, S12C Figs in spleen and circulating immune cells, respectively). This is in contrast to the mouse spleen and circulating immune cell scRNA-seq datasets, which showed nominal and nonspecific expression levels only of the adult beta and alpha hemoglobin genes (S11G–S11J and S12G–S12J Figs in spleen and circulating immune cells, respectively). It should be noted that the red-blood-cell lysis protocols we used were designed and optimized for the mouse and not the NM-R; hence, these species differences may reflect suboptimal lysis efficiency in the NM-R, possibly contributed by differences in the cell membrane components between the two species. We additionally compared the number of genes expressed in the NM-R hemoglobin-expressing clusters to the mean number of genes expressed across all NM-R clusters. In the spleen, this comparison did not indicate that these clusters express a much narrower range of genes compared with the immune cells (S11K Fig; S45 Table), unlike in the circulating immune cell data (S12K Fig; S46 Table). Taken together, we suspect that these clusters correspond to immature red blood cells, possibly in different maturation stages between the spleen and blood, and hence labeled them as erythroid cells, yet refrained from any further investigation as it fell outside the scope of this work.

In Figs 1A–1G, 3A, 3B, 3E and 3F; S2B, S3A, S3C, S4E–S4H, S5A–S5C, S10A and S10B Figs, the clusters that are represented are those obtained following the first iteration of our clustering approach. In S1E–S1G, S1L, S1M, S3B, S5D, S11 and S12 Figs, the clusters that are represented are those obtained following convergence of our clustering approach.

## Obtaining a phyletic pattern of protein-coding genes, annotated pseudogenes, and putative pseudogenes in the *LRC*, *NKC*, and *MHC-I* region

In order to detect any possible present and past members of any of the NK cell receptor gene families in the *LRC* and *NKC*, as well as of *MHC-I* genes, we followed a two-step approach. First, we used the OrthoFinder software, which finds groups of orthologous genes based on amino-acid sequence homology [35]. Specifically, OrthoFinder was used with 48 mammalian proteome sequences (S47 Table). The resulting orthogroups were then assigned to each one of the gene families based on the genome annotations in which members of these gene families were annotated. In order to include annotated pseudogenes, we augmented this step with applying a reciprocal best BLASTn [36] hit search between all pairs of the 48 mammalian transcriptomes (S47 Table), retaining only hits with an e-value <0.1 and a sequence identity >80%. Any annotated pseudogene that hit any protein-coding member, which belong to any of the gene family orthogroups, was thus additionally included in that gene family orthogroup. Second, we used each of the transcript sequences from each of the orthogroups as queries against each of the 48 mammalian genomes (S47 Table) for a SIM [37] sequence homology search, which is purposed for aligning expressed sequence tags (ESTs) against genome

sequences. Each SIM hit with at least 40% identity to the query sequence was retained. Subsequently, overlapping hits from different queries of the same gene family were merged, and the merged hit was used as query for a reverse position-specific (RPS) BLAST [38] search against conserved protein domains. Specifically, for the *KIR* and *LILR* gene families, we retained any "Ig LILR KIR like" domain hit; for the *KLRA* gene family we retained any "Ly49" domain hit; for all other NK cell receptor gene families (*KLRB-K*), we retained any "CLECT NK receptors like" domain hit; and for *MHC-I* genes, we retained any "MHC_I", "MHC_I_C", or "IgC_MHC_I_alpha3" domain hit. In all cases, any of the domain hits with either e-value $\geq 0.1$ or identity $\leq 60\%$ were filtered. We additionally required that all these SIM and RPS-BLAST hits be located on the scaffolds of either the *LRC*, *NKC*, or *MHC-I* locus of the corresponding gene queries, and for the purpose of detecting putative (i.e., unannotated) pseudogenes, annotated genes (protein-coding and pseudogenes) were ignored (S13 Table). In order to benchmark the false positive rate of this approach, for each of the gene families we counted how many of the hits covering annotated genes (protein-coding and pseudogenes) belong to the same gene family of the query (S12 Table). Because the percentage of false positives for the *LILR* and *KIR* gene families was unreasonably high, likely stemming from the confounding fact that their defining domain is shared ("Ig LILR KIR like"), we refrained from interpreting these hits as putative pseudogenes.

### Reconstructing the ancestral numbers of NK cell receptor gene families

In order to reconstruct the number of genes in each of the NK cell receptor families in each ancestral node in the phylogeny (obtained from the Ensembl database), we combined the numbers of protein-coding genes, annotated pseudogenes, and putative pseudogenes for each of the NK cell receptor gene families, except for *LILR* and *KIR*, and subsequently used that as the phylogeny tip states input for the asr_max_parsimony function of the castor [39] R [32] package, to reconstruct ancestral states using the Sankoff maximum parsimony algorithm with a cost matrix proportional to the number of genes.

### Obtaining a list of mouse–NM-R orthologous genes

To obtain mouse–NM-R one-to-one gene orthologs for the purpose of differential expression and GSEAs, we applied a reciprocal-blast approach [36]. Specifically, we used BLASTp for each of the NM-R protein sequences against all mouse protein sequences and vice versa, and the same for transcript sequences using BLASTn. In both cases, we retained only hits with e-value $<0.1$ and a sequence identity $>80\%$. Any gene for which any of its protein or transcript forms reciprocally hit back any of the protein or transcript forms of that gene was retained as a one-to-one gene ortholog (S48 Table).

### Cell type–specific responses to LPS challenge

In single-cell data (including the data generated in this study), the distribution of the number of UMIs per a given gene across the cells in a certain cluster is typically bimodal, with a left mode located at zero and a right mode located at higher than zero. Across different factors (e.g., treatments, different cell types), a change in the proportion of cells expressing the gene, i.e., a change in expression-induction, can be explicitly quantified, as well as the change in the levels of expression among the cells expressing the gene (i.e., a shift in the right modes of the distributions of the number of UMIs). To this end, in all our differential expression analyses, with the exception of the one applied for defining cluster marker genes, we fitted a hurdle generalized linear mixed model to the data using the lme4 [40] R [32] package. Specifically, effects on expression-induction were estimated by fitting a logistic mixed-effects regression model

(with a binomial family using the logit link function) by binarizing the outcome to 0 (not expressed in the cell) and 1 (expressed in the cell). Effects on expression levels were estimated by fitting a mixed-effects regression model (with a gamma family using the log link function) only to the cells in which the gene was expressed. In both cases, sample was defined as a random effect.

In order to detect intraspecies cell type–specific responses following LPS challenge, for each gene in each pair of matching cell types in the saline control and LPS challenge datasets (S49 and S50 Tables for the datasets of the spleen and circulating immune cells, respectively), we fitted the hurdle model described above to the LPS challenge and saline control data. Specifically, to estimate the effect of LPS challenge on induction of expression as well as on change of expression levels, LPS challenge was specified as a fixed categorical effect, with saline control set as baseline, and sample was specified as a random effect, as noted above. Heatmaps displaying the expression-level and expression-induction changes were limited to the set of one-to-one gene orthologs between mouse and NM-R (S48 Table) and to cell types we matched between them (S49 and S50 Tables for the datasets of the spleen and circulating immune cells, respectively). Interspecies heatmaps were generated by matching mouse and NM-R cell types across the LPS challenge and saline control conditions (S51 and S52 Tables for the datasets of the spleen and circulating immune cells datasets, respectively).

In order to perform a GSEA for the differentially expressed genes obtained using any of the fitted models, for each cell type, we ranked the genes by the $p$-value of the estimated effect and used that as an input to the fgsea function in the fgsea [41] R [32] package. We used the Hallmark gene sets of the MSigDB collection [42] for GSEAs, with the exception of the *Ltf*-high neutrophil expression-change GSEAs, in which we used the gene ontology (GO) biological pathways sets [43,44]. In all GSEAs, we performed multiple hypotheses $p$-value adjustments [45] separately for each cell type. Although not as conservative as adjusting the $p$-values across all cell types, the latter approach would suffer from being imbalanced between the two species, for which a different number of cell types was tested. To perform intra- and interspecies comparisons between proportions of cell types, we fitted a multinomial regression (similar to the mixed-effects models fitted to the expression data, except for the sample random effect) implemented by the nnet [46] R [32] package.

In order to estimate the cell trajectories of the *Ltf*-high neutrophils and neutrophils in the NM-R spleen and circulating immune cells, we applied RNA velocity [47] only to these two clusters, separately for the NM-R datasets of the spleen and circulating immune cells, where in each it was applied separately for the saline control and LPS challenge (total of four analyses).

### Comparative splenic histomorphology and fluorescent ISH

We investigated the histomorphology and expression of four genes in the spleen of four NM-Rs (two males, two females) and four C57BL/6 mice (two males, two females). Organ collection was performed as described above and spleens were immediately transferred into 10% neutral-buffered formalin. Following fixation, all samples were trimmed, routinely processed, embedded, and sectioned at 4-μm thickness in an RNase-free environment. Consecutive tissue sections were stained with hematoxylin and eosin (HE) and periodic acid–Schiff (PAS), and the expression of *Cd3e*, *Cd19*, *Cd14*, and *Rpl13a* was determined by fluorescent ISH (ViewRNA, Invitrogen, Waltham, MA). The expression of the bacterial gene *dapB* was determined as a negative control.

Samples were stained with DAPI to visualize individual cell nuclei, and the RNA ISH assay was performed according to the Invitrogen ViewRNA ISH Tissue protocol and optimized with the following conditions: heat treatment for 10 minutes at 90–95˚C and protease digestion at

1:100 dilution for 20 minutes at 40˚C. Hybridization was performed for 2 hours at 40˚C using the following species-specific probes: Mouse: *Cd14* (#VB1-3028230), *Cd3e* (#VB1-16980), *Cd19* (#VB1-3028231), *Rpl13a* (#VB1-16196); NM-R: *Cd14* (#VF1-4355532), *Cd3e* (#VF1-6000573), *Cd19* (#VF1-4359165), *Rpl13a* (#VF1-4348271) (all probes from ThermoFisher, Waltham, MA).

### ISH data analysis

Whole slide images were generated in fluorescence using Pannoramic SCAN (3D Histech, Budapest, Hungary). Image analysis data were generated by automated analysis of whole slide images using ImageDx (Reveal Biosciences, San Diego, CA), an integrated whole slide image management and automated image analysis workflow. Briefly, each image is first assessed for quality using a focus measurement followed by an accuracy check. All tissue and staining artifacts are digitally excluded from the quantification. The analysis process includes automated identification of tissue, followed by segmentation of regions of interest and then classification of each cell. The total number of cells is determined by detection of DAPI-positive nuclei. Both the number of positive staining cells and the number of positive regions within each cell are recorded. For a positive staining cell, the number of cells is provided for each number of puncta, ranging from 1 to 9 and 10 or more (i.e., ten bins).

Using the number of puncta to distinguish between cell types comes with uncertainty because one cannot exclude the possibility that cells with even a single hybridized probe (i.e., puncta = 1) are really negative for the cell type of which this probe is a marker. Clearly, this uncertainty may affect a test for determining whether mouse and NM-R spleens have different proportions of each of the three cell types targeted by each probe. In order to account for this uncertainty, for each of the ten puncta bins except for the first, we defined negative cells as those with less than the puncta bin and positive as cells with that number of puncta or higher, and fit a multinomial regression, implemented by the nnet [46] R [32] package, quantifying a species categorical effect, with mouse set as baseline. The uncertainty arising because of selection of a specific puncta bin was propagated by specifying the mean species effect across all puncta bins divided by its standard error as a z-statistic. This allowed obtaining a *p*-value, which thus served as the statistical significance of the species effect, across all puncta bins, being different from 0.

### Comparative histomorphology of peripheral blood smears

Peripheral blood smears were generated from two young female guinea pigs (approximately 2 years old) and five young nonbreeding NM-Rs (2–15 months old; 3 males and 2 females). Samples were drawn via cardiocentesis and streaked onto glass slides. Smears were air-dried and stained with Wrights-Giemsa. For each sample, the histomorphology of peripheral blood cells was assessed at 60× by a board-certified veterinary anatomic pathologist (DMI) for the presence of Foa-Kurloff cells.

### RNA isolation

One milliliter TRIzol reagent (4˚C) (Invitrogen 15596026, Waltham, MA) was added to approximately 20 mg frozen splenic tissue in a 2-mL microcentrifuge tube. A single stainless steel bead (Qiagen 69965, Hilden, Germany) was added to each tube, and tissue homogenization was performed using a Tissuelyser II (Qiagen, Hilden, Germany) (30 Hz for 2 minutes). RNA was isolated according to manufacturer's instructions (TRIzol, Invitrogen 15596026, Waltham, MA) and resuspended in RNase free water. Contaminating gDNA was removed using the Turbo DNA-free kit (Invitrogen AM1097, Waltham, MA) according to the

manufacturer's instructions. Quantification was performed using the QuBit 4 Fluorometer (Invitrogen, Waltham, MA) and the Qubit RNA XR assay (Invitrogen Q33224, Waltham, MA).

## Whole spleen RNA library preparation, sequencing, and data analysis

RNA-seq libraries from the whole splenic tissues were prepared using purified RNA isolated as described above. RNA quality and concentration were assayed using a 5200 Fragment Analyzer instrument (Agilent, Santa Clara, CA). RNA-seq libraries were prepared using the TruSeq Stranded Total RNA kit paired with the Ribo-Zero rRNA removal kit (Illumina, San Diego, CA). Libraries were sequenced on an HiSeq 4000 instrument (Illumina, San Diego, CA). Reads were mapped to the NM-R genomic references described above for the scRNA-seq data using the STAR aligner version 2.5.3a [48], applying the two-round read-mapping approach with the following parameters: outSAMprimaryFlag = "AllBestScore", outFilter MultimapNmax = "10", outFilterMismatchNoverLmax = "0.05", outFilterIntronMotifs = "RemoveNoncanonical".

Following read mapping, transcript and gene expression levels were estimated using MMSEQ [49]. Transcripts and genes that were minimally distinguishable according to the read data were collapsed using the mmcollapse utility of MMDIFF [50], and the fragments per kilobase million (FPKM) expression units were converted to TPM units. In order to test for differential expression between LPS-challenged and saline control tissues, for each species, we used MMDIFF with these two-group design matrices: # M 0, 0, 0, 0; # C 0 0, 0 0, 0 1, 0 1; # P0 1; # P1 0.5 −0.5. Samples were clustered using the heatmap.2 function from the gplots [51] R [32] package, for the purpose of constructing the genes-by-samples heatmaps for both species. GSEAs were performed for each species, for which the inputs to the fgsea function in the fgsea [41] R [32] package were the genes ranked in descending order according to the Bayes Factor and posterior probability reported by MMDIFF, and the gene sets were the Hallmark gene sets of the MSigDB collection [42].

## Supporting information

**S1 Fig. Overview of scRNA-seq and transcriptional maps of mouse and NM-R spleens. (A)** Schematic view showing the workflow in which single-cell suspensions are derived from four (two males [M], two females [F]) C57BL/6 mouse spleens. **(B)** Bar chart showing the number of cells sequenced in duplicate from each of the four mouse spleens (see S43 Table for underlying data). **(C)** Violin plot showing the number of genes sequenced per cell from each of the four mouse spleens (see S1 Table for underlying data). **(D)** Violin plot showing the number of UMIs sequenced per cell from each of the four mouse spleens (see S1 Table for underlying data). **(E)** UMAP projection of the four-mouse spleen scRNA-seq data, for which each point is a cell color-coded by its converged-cluster assignment and annotated cell type. **(F)** Gene-by-cell expression-level heatmap of the four-mouse spleen scRNA-seq data, in which selected marker genes are listed to the left and cells are faceted by their converged cluster assignment. **(G)** Stacked bar chart showing the proportion (%) of cells from each of the four mouse duplicate spleen samples assigned to each of the converged clusters (see S1 Table for underlying data). Samples are color-coded and duplicates are shade-coded. **(H)** Schematic view showing the workflow in which single-cell suspensions are derived from four (two males [M], two females [F]) NM-R spleens. **(I)** Bar chart showing the number of cells sequenced in duplicate from each of the four NM-R spleens (see S43 Table for underlying data). **(J)** Violin plot showing the number of genes sequenced per cell from each of the four NM-R spleens (see S2 Table for underlying data). **(K)** Violin plot showing the number of UMIs sequenced per cell from

each of the four NM-R spleens (see S2 Table for underlying data). **(L)** UMAP projection of the four NM-R spleen scRNA-seq data, for which each point is a cell color-coded by its converged-cluster assignment and annotated cell type. **(M)** Gene-by-cell expression-level heatmap of the four NM-R spleen scRNA-seq data, in which selected marker genes are listed to the left and cells are faceted by their converged cluster assignment. **(N)** Stacked bar chart showing the proportion (%) of cells from each of the four NM-R duplicate spleen samples assigned to each of the converged clusters (see S2 Table for underlying data). Samples are color-coded and duplicates are shade-coded. DC, dendritic cell; MZ, marginal zone; NK, natural killer; NKT, natural killer T; NM-R, naked mole-rat; RP, red pulp; scRNA-seq, single-cell RNA-sequencing; UMAP, uniform manifold approximation and projection; UMI, unique molecular identifier. (TIF)

**S2 Fig. Differences in splenic microarchitecture and gene expression as measured by ISH recapitulate the differences observed in scRNA-seq data. (A)** Representative images of HE-stained consecutive sections of mouse (upper panel) and NM-R (lower panel) spleens shown at 5× (left panels) and 20× magnifications (right panels), scale bar = 100 μm. Mouse and NM-R spleens show major differences in splenic microanatomy, with the NM-R having a comparatively reduced white-pulp compartment and a larger red-pulp compartment with a greater number of fibromuscular trabeculae that connect to the capsule and provide structural support and contractility to the spleen. Within the white pulp, the marginal zone and follicles of the NM-R (which comprise B cells) are readily identifiable. By contrast, the PALS (which comprises the T cell–rich compartment in other species) is less prominent in the NM-R than in the mouse. **(B)** UMAP projections of the mouse (upper panels) and NM-R (lower panels) spleen scRNA-seq data color-coded by the expression levels of myeloid and lymphoid lineage marker genes: *Cd14* (myeloid marker), *Cd19* (B cell marker, lymphoid), and *Cd3e* (T cell marker, lymphoid). **(C)** Representative consecutive sections of mouse (upper panel) and NM-R (lower panel) spleens showing results from ISH staining for *Cd14*, *Cd19*, and *Cd3e*. Positive expression of the marker genes was visualized with a TRITC filter (yellow) and nuclei visualized using DAPI staining (blue). Magnification, 20×. Scale bar = 100 μm. Expression of the housekeeping gene *Rpl13a* and the bacterial-specific *dapB* gene were included as positive and negative controls, respectively (Materials and methods). **(D)** Quantification of the ISH staining of *Cd14*, *Cd19*, and *Cd3e* in mouse (gray bars) and NM-R (pink bars). Bars represent the proportion of positive staining cells ($n = 4$ biological replicates) and error bars represent the uncertainty associated with the number of puncta used as a cutoff for defining a cell as positive for a marker gene (Materials and methods and S5 Table for underlying data). Asterisks mark statistically significant (adjusted $p < 0.05$) differences in cell type proportions. Consistent with our findings from the scRNA-seq data, ISH showed a significantly higher level of *Cd14* in the NM-R compared with the mouse, with staining restricted to the red-pulp regions in both species. However, the NM-R has significantly lower abundance of *Cd19*, with positive staining cells localized to the follicular regions of the white pulp. Compared with the dramatic differences in the abundance of *Cd14* and *Cd19*, the abundance of *Cd3e* is similar between the two species, which is in line with our scRNA-seq data. Despite their similar abundance, the distribution of *Cd3e* positive staining cells is markedly different between the two species, with *Cd3e* expression in the mouse limited to the PALS, while that of the NM-R is stochastically distributed throughout both the red- and white-pulp regions, with only minimal aggregation in the PALS. Thus, direct examination of splenic tissue from mouse and NM-R highlights major differences in microanatomy between the two species and supports our results from the scRNA-seq data with respect to the proportions of the major immune cell lineages in the two species. APC, antigen-presenting cell; DC, dendritic cell; HE, hematoxylin and eosin; ISH, in situ

hybridization; NK, natural killer; NM-R, naked mole-rat; PALS, periarteriolar lymphoid sheath; scRNA-seq, single-cell RNA-sequencing; TRITC, tetramethylrhodamine; UMAP, uniform manifold approximation and projection.
(TIF)

**S3 Fig. Expression of canonical NK cell marker genes in the NM-R spleen is confined to T-cell subsets and mast cells. (A)** UMAP projections of the four NM-R spleens scRNA-seq data color-coded by the expression levels of three canonical NK cell marker genes: *Nkg7*, *Gzma*, and *Ccl5*. **(B)** Violin plots showing the expression levels of the CD3 subunit genes: *Cd3d*, *Cd3e*, *Cd3g*, *Cd247*, and of *Cd8a* in the mouse T-cell and NK cell converged clusters (left panel) and NM-R T-cell converged clusters (center panel). The side panels show effect sizes for expression-level changes and expression-induction changes, where in the mouse, the naïve CD8 T cells are used as baseline and in the NM-R, the naïve T cells are used as baseline (see S6 Table for underlying data). Asterisks mark adjusted $p < 0.05$. **(C)** UMAP projections of the four NM-R spleens scRNA-seq data color-coded by the expression levels of three mast cells marker genes: *Tpsab1*, *Cma1*, and *Fcer1a*. CD3, cluster of differentiation 3; Mem, memory; NK, natural killer; NKT, natural killer T; NM-R, naked mole-rat; scRNA-seq, single-cell RNA-sequencing; UMAP, uniform manifold approximation and projection.
(TIF)

**S4 Fig. scRNA-seq of mouse and NM-R circulating immune cells confirms absence of canonical NK cells in the NM-R. (A)** Schematic view showing the workflow in which single-cell suspensions are derived from circulating immune cells of four (two males [M], two females [F]) mice (upper panel, gray) and four NM-Rs (two males [M], two females [F]) (lower panel, pink). **(B)** Bar charts showing the number of cells sequenced from each of the four mice (upper panel, gray) and three NM-Rs (one sample was lost due to a microfluidic failure during the emulsion generation; lower panel, pink) (see S43 Table for underlying data). **(C)** Violin plots showing the number of genes sequenced per cell from each of the four mice (upper panel, gray; see S7 Table for underlying data) and three NM-Rs (lower panel, pink; see S8 Table for underlying data). **(D)** Violin plots showing the number of UMIs sequenced per cell from each of the four mice (upper panel, gray; see S7 Table for underlying data) and three NM-Rs (lower panel, pink; see S8 Table for underlying data). UMAP projections of the four mouse **(E)** and three NM-R **(F)** circulating immune scRNA-seq datasets. Each point is a cell color-coded by its first iteration cluster assignment and annotated cell type. The proportions of each cell type are shown in the central bar chart. Gene-by-cell expression-level heatmaps of the four mouse **(G)** and three NM-R **(H)** circulating immune scRNA-seq datasets. Selected marker genes are listed to the left and cells are faceted by their first iteration cluster assignment. DC, dendritic cell; NK, natural killer; NM-R, naked mole-rat; scRNA-seq, single-cell RNA-sequencing; UMAP, uniform manifold approximation and projection; UMI, unique molecular identifier.
(TIF)

**S5 Fig. Absence of canonical NK cells in the NM-R cannot be explained by annotation deficiencies in the NM-R genome.** One possibility for the absence of NK cells in the NM-R may be that, due to incomplete annotation of NK cell marker genes in the NM-R, genome reads generated from these marker genes would be filtered as a result of not mapping back to the annotated genome. To test this possibility, we simulated a lack of NK cell marker genes in the mouse by selectively eliminating the mouse NK cell marker genes (51 and 43 genes in the spleen and circulating immune cells; S3 and S9 Tables, respectively) and subsequently re-clustered the mouse data (done both for the spleen data as well as for the circulating immune

cells). **(A)** UMAP projection of the four mouse spleen scRNA-seq data color-coded by the first iteration clusters achieved using the full set of genes. **(B)** UMAP projection of the four mouse spleen scRNA-seq data color-coded by the first iteration clusters achieved using the set of genes without the mouse spleen NK cell marker genes. Clusters are labeled by their indices and by the percentage of their cells that map to the cluster/cell type they correspond to in **(A)**. **(C)** UMAP projection of the four mouse circulating immune scRNA-seq data color-coded by the first iteration clusters achieved using the full set of genes. **(D)** UMAP projection of the four mouse circulating immune scRNA-seq data color-coded by the converged clusters achieved using the set of genes without the mouse circulating NK cell marker genes. Clusters are labeled by their indices and by the percentage of their cells that map to the cluster/cell type they correspond to in **(C)**. APC, antigen-presenting cell; NK, natural killer; NM-R, naked mole-rat; scRNA-seq, single-cell RNA-sequencing; UMAP, uniform manifold approximation and projection.
(TIF)

**S6 Fig. Gene expression and histological support for the lack of NM-R canonical NK cells.** **(A)** The amino acid sequence of the NM-R *Ly49* gene (*LOC101714034*; a mouse *Klra1* ortholog) color-coded by its predicted functional domains: extracellular region (green), intracellular region (blue), and transmembrane region (red). The immunoreceptor tyrosine inhibitory motif (ITIM) is shown in pink. **(B)** UMAP projections of the NM-R spleen scRNA-seq data color-coded by clusters (left) and by the expression levels of *LOC101714034* (right), showing its low expression levels in CD8 T cells. **(C)** UMAP projections of the mouse spleen scRNA-seq data color-coded by clusters (left) and by the expression levels of *Klra1* (right), showing its high expression levels in CD8 T cells and NK cells. **(D)** Wrights-Giemsa stained peripheral blood smear (magnification = 60×; scale bar = 25 μm) from a young female guinea pig, with arrows marking Foa-Kurloff cells characterized by a large cuniform cytoplasmic inclusion body, along with two polymorphonuclear neutrophils (top left and center) and a lymphocyte (bottom). **(E-I)** Wrights-Giemsa stained peripheral blood smears (magnification = 60×; scale bar = 25 μm) from five young NM-Rs ([**E**] 2-month-old male, [**F**] 7-month-old male, [**G**] 15-month-old female, [**H**] 11-month-old male, [**I**] 7-month-old female) showing mainly polymorphonuclear neutrophils and clumped platelets, along with a few lymphocytes, monocytes, and eosinophils with granular cytoplasm, where none of these leukocytes show an overt Foa-Kurloff cell morphology. CD8, cluster of differentiation 8; DC, dendritic cell; NK, natural killer; NKT, natural killer T; NM-R, naked mole-rat; pDC, plasmacytoid dendritic cell; RP, red pulp; scRNA-seq, single-cell RNA-sequencing; UMAP, uniform manifold approximation and projection.
(TIF)

**S7 Fig. Reconstructing the ancestral counts of NK cell receptor gene families. (A)** Stacked bar chart showing the benchmarked percentage of false positive putative NK cell receptor and *MHC-I* pseudogenes (Materials and methods and S12 Table for underlying data) for each gene family (bars), color-coded by the genome of the search species. Labeled on top of each bar is (number of false positives)/(number of true + false positives). Horizontal black dashed line marks the 5% false positive rate. While the high false positive rate in the *LILR* and *KIR* gene families is driven by many of the search genomes, for the remaining gene families the main drivers of the false positive rate are the platypus and donkey search genomes (orange and purple shades, respectively). **(B)** A similar stacked bar chart as **(A)** with the platypus and donkey search genomes removed, showing that in the gene families other than *LILR* and *KIR*, all gene families except for *KLRK* have either a single false positive or a false positive rate <5%. **(C-L)** Phylogenies of the 48 mammalian genomes; each corresponds to one of the NK cell receptor

gene families within the *NKC* (except for *KLRA*): *KLRG* (**C**), *KLRB* (**D**), *KLRE* (**E**), *KLRF* (**F**), *KLRD* (**G**), *KLRK* (**H**), *KLRC* (**I**), *KLRJ* (**J**), *KLRI* (**K**), and *KLRH* (**L**). Color shades correspond to the numbers of genes. Because the ancestral reconstruction is not deterministic, ancestral counts are represented as pie charts corresponding to the probability of each count. *KIR*, killer cell immunoglobulin-like receptor; *KLR*, killer cell lectin-like receptor; *LILR*, leukocyte immunoglobulin-like receptor; *MHC-I*, major histocompatibility complex class I; NK, natural killer; *NKC*, natural killer cell receptor complex.
(TIF)

**S8 Fig. Whole-spleen RNA sequencing of mice and NM-Rs following LPS challenge. (A)** Genes-by-samples heatmap showing the scaled ln(TPM) expression levels of genes in spleens from the two saline control and two LPS-challenged mice. Selected LPS up-regulated genes are shown to the left. **(B)** Genes-by-samples heatmap showing the scaled ln(TPM) expression levels in spleens from the two saline control and two LPS-challenged NM-Rs. Selected LPS up-regulated genes are shown to the right. **(C)** Volcano plot showing the posterior probability of the estimated effect size (ln(LPS/saline) ln(TPM) fold-change) being different from zero (y-axis) versus the estimated effect size (x-axis) in the mouse data. Each point is a gene, and the color code follows the posterior probability gradient. **(D)** Volcano plot showing the posterior probability of the estimated effect size (ln(LPS/saline) ln(TPM) fold-change) being different from zero (y-axis) versus the estimated effect size (x-axis) in the NM-R data. Each point is a gene, and the color code follows the posterior probability gradient. **(E)** Bar chart showing the Hallmark gene sets enriched in genes with strong expression changes following LPS challenge in mice (see S16 Table for underlying data). The x-axis reports the $\log_{10}$-adjusted *p*-value (*q*-value) of the GSEA, signed by the direction of the effect (up- and down-regulation in LPS relative to saline are represented as follows: positive is yellow and negative is cyan, respectively). Vertical dashed lines represent adjusted *p* = 0.1. **(F)** Bar chart showing the Hallmark gene sets enriched in genes with strong expression changes following LPS challenge in NM-Rs (see S17 Table for underlying data). The x-axis reports the $\log_{10}$-adjusted *p*-value (*q*-value) of the GSEA, signed by the direction of the effect (up- and down-regulation in LPS relative to saline are represented as follows: positive is yellow and negative is cyan, respectively). Vertical dashed lines represent adjusted *p* = 0.1. GSEA, gene set enrichment analysis; LPS, lipopolysaccharide; NM-R, naked mole-rat; TPM, transcripts per million.
(TIF)

**S9 Fig. Cell-specific responses to LPS challenge in the NM-R and mouse spleen.** Marked transcriptional responses were evident following LPS challenge for each cell type compared between the LPS-challenged and saline control animals in each of the species. As previously described by Shalek and colleagues [52], these LPS-responsive transcriptional changes were driven by expression-level change (a change in the level of the expression among the cells that express a given gene in both conditions) and/or expression-induction change (a change in the proportion of cells expressing a certain gene in both conditions), illustrated here by the transcriptional changes of *Nfkbia*. **(A)** Selected density plots showing the expression distribution of *Nfkbia* from saline control (orange) and LPS-challenged (red) mice across the cells of five representative cell types. Insets show the estimated LPS-challenge expression-level change (Exp.) and expression-induction change (Ind.) effect sizes (LPS relative to saline) (asterisks mark adjusted *p* < 0.05). **(B)** Selected density plots showing the expression distribution of *Nfkbia* from saline control (orange) and LPS-challenged (red) NM-Rs across the cells of five representative cell types. Insets show the estimated LPS-challenge expression-level change (Exp.) and expression-induction change (Ind.) effect sizes (LPS relative to saline) (asterisks mark adjusted *p* < 0.05). **(C)** Heatmap showing the estimated LPS-challenge effect statistics

for expression-level change. Selected marker genes are shown to the left. **(D)** Heatmap showing the estimated LPS-challenge effect statistics for expression-induction change. Selected marker genes are shown to the right. **(E)** Bar chart showing the GSEAs of the intraspecies LPS-challenge effect on expression-level change in mouse NK cells (see S26 Table for underlying data). The x-axis is the GSEA $\log_{10}$-adjusted $p$-values ($q$-values), signed and color-coded by the direction of the effect (up- and down-regulation in LPS-challenge relative to saline control are represented as yellow and cyan, respectively). Black vertical dashed lines mark $q = 0.1$. **(F)** Heatmap showing GSEAs of the intraspecies LPS-challenge effect on expression-induction change in mice across the listed cell types (see S27 Table for underlying data). The GSEA $\log_{10}$-adjusted $p$-values ($q$-values) are color-coded by the direction of the effect (up- and down-regulation in LPS relative to saline: yellow and cyan, respectively) and shaded by the effect $q$-value. Gene sets with $q < 0.1$ are indicated with an asterisk. **(G)** Heatmap showing GSEAs of the intraspecies LPS-challenge on expression-induction change in NM-Rs across the listed cell types (see S28 Table for underlying data). The GSEA $\log_{10}$-adjusted $p$-values ($q$-values) are color-coded by the direction of the effect (up- and down-regulation in LPS relative to saline are represented as yellow and cyan, respectively) and shaded by the effect $q$-value. Gene sets with $q < 0.1$ are indicated with an asterisk. APC, antigen-presenting cell; B, B cell; GSEA, gene set enrichment analysis; LPS, lipopolysaccharide; Mono, monocyte; Neut, neutrophil; *Nfkbia*, nuclear factor kappa-light-chain-enhancer of activated B cells inhibitor alpha; NK, natural killer; NM-R, naked mole-rat; T, T cell.
(TIF)

**S10 Fig. Circulating immune scRNA-seq reveals that NM-Rs have an LPS-responsive cell subset not found in mice.** UMAP projections showing the clusters of the circulating immune cells from the two saline control (left panel) and two LPS-challenged (right panel) mice **(A)** and NM-Rs **(B)**. Each point is a cell color-coded by its cluster assignment and annotated cell type. Stacked bar charts showing the proportions (%) of each cell type in the saline control (top) and LPS-challenged (bottom) mouse **(C)** and NM-R **(D)** circulating immune cells. Similar to the spleen saline control and LPS challenge data (S9 Fig), marked transcriptional responses were evident following LPS challenge for each cell type compared between the LPS-challenged and saline control animals in each of the species. And these changes were again driven by expression-level changes and/or expression-induction changes, illustrated here by the transcriptional changes of *Nfkbia*. **(E)** Selected density plots showing the expression distribution of *Nfkbia* from saline control (orange) and LPS-challenged (red) mice across the cells of five representative cell types. Insets show the estimated LPS-challenge expression-level change (Exp.) and expression-induction change (Ind.) effect sizes (LPS relative to saline) (asterisks mark adjusted $p < 0.05$). **(F)** Selected density plots showing the expression distribution of *Nfkbia* from saline control (orange) and LPS-challenged (red) NM-Rs across the cells of five representative cell types. Insets show the estimated LPS-challenge expression-level change (Exp.) and expression-induction change (Ind.) effect sizes (LPS relative to saline) (asterisks mark adjusted $q < 0.05$). **(G)** Heatmap showing the estimated LPS-challenge effect statistics for expression-level change. Selected marker genes are shown to the left. **(H)** Heatmap showing the estimated LPS-challenge effect statistics for expression-induction change. Selected marker genes are shown to the right. **(I)** Bar chart showing the GSEAs of the intraspecies LPS-challenge effect on expression-level change in mouse NK cells (see S39 Table for underlying data). The x-axis is the GSEA $\log_{10}$-adjusted $p$-values ($q$-values) signed and color-coded by the direction of the effect (up- and down-regulation in LPS-challenge relative to saline control are represented by yellow and cyan, respectively). Black vertical dashed lines mark $q = 0.1$. **(J)** Heatmap showing GSEAs of the intraspecies LPS-challenge effect on

expression-induction change in mice across the listed cell types (see S40 Table for underlying data). The GSEA $\log_{10}$-adjusted $p$-values ($q$-values) are color-coded by the direction of the effect (up- and down-regulation in LPS relative to saline are represented as yellow and cyan, respectively) and shaded by the effect $q$-value. Gene sets with $q < 0.1$ are indicated with an asterisk. **(K)** Heatmap showing GSEAs of the intraspecies LPS-challenge on expression-induction change in NM-Rs across the listed cell types (see S41 Table for underlying data). The GSEA $\log_{10}$-adjusted $p$-values ($q$-values) are color-coded by the direction of the effect (up- and down-regulation in LPS relative to saline are represented by yellow and cyan, respectively) and shaded by the effect $q$-value. Gene sets with $q < 0.1$ are indicated with an asterisk. UMAP projections showing cell clusters representing *Ltf*-high neutrophils (pink) and neutrophils (yellow) from saline control **(L)** and LPS-challenged **(M)** NM-R circulating immune cells. Overlaid on each cluster are arrows that represent the cell trajectory (Materials and methods). **(N)** Bar chart showing GSEA comparing circulating *Ltf*-high neutrophils to neutrophils in LPS-challenged NM-Rs (see S42 Table for underlying data). The x-axis is the GSEA $\log_{10}$-adjusted $p$-values ($q$-values), signed and color-coded by the direction of the effect (up- and down-regulation in *Ltf*-high neutrophils versus neutrophils are represented by yellow and cyan, respectively). Black vertical dashed lines mark adjusted $p = 0.1$. B, B cell; Ery, erythroid cell; GSEA, gene set enrichment analysis; LPS, lipopolysaccharide; Mono, monocyte; Neut, neutrophil; *Nfkbia*, nuclear factor kappa-light-chain-enhancer of activated B cells inhibitor alpha; NK, natural killer; NM-R, naked mole-rat; RBC, red blood cell; scRNA-seq, single-cell RNA-sequencing; UMAP, uniform manifold approximation and projection. (TIF)

**S11 Fig. Hemoglobin gene expression in the spleen scRNA-seq data.** UMAP projections of the clusters of the NM-R and mouse spleen datasets color-coded by the expression levels of **(A)** NM-R beta hemoglobin, **(B)** NM-R gamma hemoglobin, **(C)** NM-R epsilon hemoglobin, **(D)** NM-R alpha hemoglobin, **(E)** NM-R theta hemoglobin, **(F)** NM-R zeta hemoglobin, **(G)** mouse beta-s hemoglobin, **(H)** mouse beta-t hemoglobin, **(I)** mouse alpha-1 hemoglobin, and **(J)** mouse alpha-2 hemoglobin. **(K)** Violin plot showing the numbers of genes expressed in each of the NM-R spleen converged clusters (right panel) and a plot showing the effect sizes comparing the number of genes in each of the NM-R converged clusters to the mean across all converged clusters (right panel; see S45 Table for the underlying data). Asterisks mark adjusted $p < 0.05$. DC, dendritic cell; NM-R, naked mole-rat; RP, red pulp; scRNA-seq, single-cell RNA-sequencing; UMAP, uniform manifold approximation and projection. (TIF)

**S12 Fig. Hemoglobin gene expression in the immune circulating scRNA-seq data.** UMAP projections of the clusters of the NM-R and mouse circulating immune cell datasets color-coded by the expression levels of **(A)** NM-R beta hemoglobin, **(B)** NM-R gamma hemoglobin, **(C)** NM-R epsilon hemoglobin, **(D)** NM-R alpha hemoglobin, **(E)** NM-R theta hemoglobin, **(F)** NM-R zeta hemoglobin, **(G)** mouse beta-s hemoglobin, **(H)** mouse beta-t hemoglobin, **(I)** mouse alpha-1 hemoglobin, and **(J)** mouse alpha-2 hemoglobin. **(K)** Violin plot showing the numbers of genes expressed in each of the converged clusters of the NM-R circulating immune cells data (right panel) and a plot showing the effect sizes comparing the number of genes in each of the NM-R converged clusters to the mean across all converged clusters (right panel; see S46 Table for the underlying data). Asterisks mark adjusted $p < 0.05$. DC, dendritic cell; NM-R, naked mole-rat; scRNA-seq, single-cell RNA-sequencing; UMAP, uniform manifold approximation and projection. (TIF)

**S1 Table. Mouse spleen scRNA-seq clustering data.** For each mouse spleen cell in our scRNA-seq data, the corresponding sample, number of captured UMIs, number of captured genes, first-iteration and convergence cluster and cell type assignments are provided. scRNA-seq, single-cell RNA-sequencing; UMI, unique molecular identifier.
(CSV)

**S2 Table. NM-R spleen scRNA-seq clustering data.** For each NM-R spleen cell in our scRNA-seq data, the corresponding sample, number of captured UMIs, number of captured genes, first-iteration and convergence cluster and cell type assignments are provided. NM-R, naked mole-rat; scRNA-seq, single-cell RNA-sequencing; UMI, unique molecular identifier.
(CSV)

**S3 Table. Mouse spleen scRNA-seq cluster marker genes.** For each marker gene in each of the converged clusters of the mouse spleen scRNA-seq data, provided are its Ensembl gene ID, gene symbol, gene description, gene biotype, and converged cluster and cell type. scRNA-seq, single-cell RNA-sequencing.
(CSV)

**S4 Table. NM-R spleen scRNA-seq cluster marker genes.** For each marker gene in each of the converged clusters of the NM-R spleen scRNA-seq data, provided are its RefSeq gene ID, gene symbol, and converged cluster and cell type. NM-R, naked mole-rat; scRNA-seq, single-cell RNA-sequencing.
(CSV)

**S5 Table. Spleen in situ data.** For each mouse and NM-R sample, provided for each probe gene are the species, sample, number of puncta bin, cell count, and cell area. NM-R, naked mole-rat.
(CSV)

**S6 Table. Differential expression of the CD3 subunit genes: *Cd3d, Cd3e, Cd3g, Cd247*, and of *Cd8a* among mouse and NM-R clusters.** For each differential expression contrast (for each gene in each converged cluster), provided are the gene symbol, converged cluster cell type, species, test (differential induction or expression), effect size, standard error of effect size, *p*-value, and adjusted *p*-value (*q*-value). CD3, cluster of differentiation 3; NM-R, naked mole-rat.
(CSV)

**S7 Table. Mouse circulating immune scRNA-seq clustering data.** For each mouse circulating immune cell in our scRNA-seq data, the corresponding sample, number of captured UMIs, number of captured genes, first-iteration and convergence cluster and cell type assignments are provided. scRNA-seq, single-cell RNA-sequencing; UMI, unique molecular identifier.
(CSV)

**S8 Table. NM-R circulating immune scRNA-seq clustering data.** For each NM-R circulating immune cell in our scRNA-seq data, the corresponding sample, number of captured UMIs, number of captured genes, first-iteration and convergence cluster and cell type assignments are provided. NM-R, naked mole-rat; scRNA-seq, single-cell RNA-sequencing; UMI, unique molecular identifier.
(CSV)

**S9 Table. Mouse circulating immune scRNA-seq cluster marker genes.** For each marker gene in each of the converged clusters of the mouse circulating immune scRNA-seq data, provided are its Ensembl gene ID, gene symbol, gene description, gene biotype, and converged

cluster and cell type. scRNA-seq, single-cell RNA-sequencing.
(CSV)

**S10 Table. NM-R circulating immune scRNA-seq cluster marker genes.** For each marker gene in each of the converged clusters of the NM-R circulating immune scRNA-seq data, provided are its RefSeq gene ID, gene symbol, and converged cluster and cell type. NM-R, naked mole-rat; scRNA-seq, single-cell RNA-sequencing.
(CSV)

**S11 Table. Data corresponding to Fig 2A–2C showing the genes in the *LRC, NKC,* and *MHC-I* region of the human, mouse, and NM-R genomes.** For each species, for each gene (protein coding, annotated, and putative pseudogenes) provided are the gene ID (Ensembl for all species except NM-R, for which it is RefSeq), gene symbol, gene biotype, genome scaffold name, start site, end site, and gene family. *LRC*, leukocyte receptor complex; *MHC-I*, major histocompatibility complex class I; *NKC*, natural killer cell receptor complex; NM-R, naked mole-rat.
(CSV)

**S12 Table. Benchmarking the detection of putative pseudogenes in the *LRC, NKC,* and *MHC-I* region.** For each species, for each gene family, provided are the species and gene ID (Ensembl for all species except NM-R, for which it is RefSeq) of the genes used as query, the hit gene ID, symbol, and family. *LRC*, leukocyte receptor complex; *MHC-I*, major histocompatibility complex class I; *NKC*, natural killer cell receptor complex; NM-R, naked mole-rat.
(CSV)

**S13 Table. Phyletic pattern data of the protein-coding genes, annotated pseudogenes, and putative pseudogenes in the *LRC, NKC,* and *MHC-I* region.** For each species, for each gene (protein coding, annotated and putative pseudogene) provided are the gene ID (Ensembl for all species except NM-R, for which it is RefSeq), gene symbol, gene family, biotype, genome scaffold name, start and end site. *LRC*, leukocyte receptor complex; *MHC-I*, major histocompatibility complex class I; *NKC*, natural killer cell receptor complex; NM-R, naked mole-rat.
(CSV)

**S14 Table. Mouse whole spleen RNA-sequencing LPS challenge versus saline control differential expression.** For each expressed gene provided are its Ensembl ID, symbol, description, biotype, Entrez ID; estimated ln fold-change, its posterior probability of being different from zero, and a Bayes factor of the differential expression model versus a null model; the ln (TPM) of each sample, the mean and standard deviation of the ln(TPM) across the samples of each condition, and the number of uniquely mapped reads in each sample. LPS, lipopolysaccharide.
(CSV)

**S15 Table. NM-R whole spleen RNA-sequencing LPS challenge versus saline control differential expression.** For each expressed gene provided are its RefSeq ID, symbol; estimated ln fold-change, its posterior probability of being different from zero, and a Bayes factor of the differential expression model versus a null model; the ln(TPM) of each sample, the mean and standard deviation of the ln(TPM) across the samples of each condition, and the number of uniquely mapped reads in each sample. LPS, lipopolysaccharide; NM-R, naked mole-rat; TPM, transcripts per million.
(CSV)

**S16 Table. Mouse whole spleen RNA-sequencing LPS challenge versus saline control GSEA.** For each enriched gene set provided are its category description, *p*-value, adjusted *p*-value (*q*-value), test statistic, enrichment direction (up or down), and the Entrez IDs and symbols of the leading-edge genes. GSEA, gene set enrichment analysis; LPS, lipopolysaccharide.
(CSV)

**S17 Table. NM-R whole spleen RNA-sequencing LPS challenge versus saline control GSEA.** For each enriched gene set provided are its category description, *p*-value, adjusted *p*-value (*q*-value), test statistic, enrichment direction (up or down), and the Entrez IDs and symbols of the leading-edge genes. GSEA, gene set enrichment analysis; LPS, lipopolysaccharide; NM-R, naked mole-rat.
(CSV)

**S18 Table. Mouse saline control spleen scRNA-seq clustering data.** For each mouse saline control spleen cell in our scRNA-seq data, the corresponding sample, number of captured UMIs, number of captured genes, first-iteration and convergence cluster and cell type assignments are provided. scRNA-seq, single-cell RNA-sequencing; UMI, unique molecular identifier.
(CSV)

**S19 Table. Mouse LPS challenge spleen scRNA-seq clustering data.** For each mouse LPS challenge spleen cell in our scRNA-seq data, the corresponding sample, number of captured UMIs, number of captured genes, first-iteration and convergence cluster and cell type assignments are provided. LPS, lipopolysaccharide; scRNA-seq, single-cell RNA-sequencing; UMI, unique molecular identifier.
(CSV)

**S20 Table. NM-R saline control spleen scRNA-seq clustering data.** For each NM-R spleen saline control cell in our scRNA-seq data, the corresponding sample, number of captured UMIs, number of captured genes, first-iteration and convergence cluster and cell type assignments are provided. LPS, lipopolysaccharide; NM-R, naked mole-rat; scRNA-seq, single-cell RNA-sequencing; UMI, unique molecular identifier.
(CSV)

**S21 Table. NM-R LPS challenge spleen scRNA-seq clustering data.** For each NM-R spleen LPS challenge cell in our scRNA-seq data, the corresponding sample, number of captured UMIs, number of captured genes, first-iteration and convergence cluster and cell type assignments are provided. LPS, lipopolysaccharide; NM-R, naked mole-rat; scRNA-seq, single-cell RNA-sequencing; UMI, unique molecular identifier.
(CSV)

**S22 Table. Mouse saline control spleen scRNA-seq cluster marker genes.** For each marker gene in each of the converged clusters of the mouse spleen saline control scRNA-seq data, provided are its Ensembl gene ID, gene symbol, gene description, gene biotype, and converged cluster and cell type. scRNA-seq, single-cell RNA-sequencing.
(CSV)

**S23 Table. Mouse LPS challenge spleen scRNA-seq cluster marker genes.** For each marker gene in each of the converged clusters of the mouse spleen LPS challenge scRNA-seq data, provided are its Ensembl gene ID, gene symbol, gene description, gene biotype, and converged cluster and cell type. LPS, lipopolysaccharide; scRNA-seq, single-cell RNA-sequencing.
(CSV)

**S24 Table. NM-R saline control spleen scRNA-seq cluster marker genes.** For each marker gene in each of the converged clusters of the NM-R spleen saline control scRNA-seq data, provided are its RefSeq gene ID, gene symbol, and converged cluster and cell type. NM-R, naked mole-rat; scRNA-seq, single-cell RNA-sequencing.
(CSV)

**S25 Table. NM-R LPS challenge spleen scRNA-seq cluster marker genes.** For each marker gene in each of the converged clusters of the NM-R spleen LPS challenge scRNA-seq data, provided are its RefSeq gene ID, gene symbol, and converged cluster and cell type. LPS, lipopolysaccharide; NM-R, naked mole-rat; scRNA-seq, single-cell RNA-sequencing.
(CSV)

**S26 Table. Mouse LPS challenge versus saline control spleen scRNA-seq expression change GSEA.** For each cluster, for each enriched gene set, provided are its category description, $p$-value, adjusted $p$-value ($q$-value), test statistic, enrichment direction (up or down), and the Entrez IDs and symbols of the leading-edge genes. GSEA, gene set enrichment analysis; LPS, lipopolysaccharide; scRNA-seq, single-cell RNA-sequencing.
(CSV)

**S27 Table. Mouse LPS challenge versus saline control spleen scRNA-seq expression induction GSEA.** For each cluster, for each enriched gene set, provided are its category description, $p$-value, adjusted $p$-value ($q$-value), test statistic, enrichment direction (on or off), and the Entrez IDs and symbols of the leading-edge genes. GSEA, gene set enrichment analysis; LPS, lipopolysaccharide; scRNA-seq, single-cell RNA-sequencing.
(CSV)

**S28 Table. NM-R LPS challenge versus saline control spleen scRNA-seq expression induction GSEA.** For each cluster, for each enriched gene set, provided are its category description, $p$-value, adjusted $p$-value ($q$-value), test statistic, enrichment direction (on or off), and the Entrez IDs and symbols of the leading-edge genes. GSEA, gene set enrichment analysis; LPS, lipopolysaccharide; NM-R, naked mole-rat; scRNA-seq, single-cell RNA-sequencing.
(CSV)

**S29 Table. NM-R spleen saline control *Ltf*-high neutrophils versus neutrophils scRNA-seq expression change GSEA.** For each cluster, for each enriched gene set, provided are its category description, $p$-value, adjusted $p$-value ($q$-value), test statistic, enrichment direction (on or off), and the Entrez IDs and symbols of the leading-edge genes. GSEA, gene set enrichment analysis; NM-R, naked mole-rat; scRNA-seq, single-cell RNA-sequencing.
(CSV)

**S30 Table. NM-R spleen LPS challenge *Ltf*-high neutrophils versus neutrophils scRNA-seq expression change GSEA.** For each cluster, for each enriched gene set, provided are its category description, $p$-value, adjusted $p$-value ($q$-value), test statistic, enrichment direction (on or off), and the Entrez IDs and symbols of the leading-edge genes. GSEA, gene set enrichment analysis; LPS, lipopolysaccharide; NM-R, naked mole-rat; scRNA-seq, single-cell RNA-sequencing.
(CSV)

**S31 Table. Mouse saline control circulating immune scRNA-seq clustering data.** For each mouse saline control circulating immune cell in our scRNA-seq data, the corresponding sample, number of captured UMIs, number of captured genes, first-iteration and convergence cluster and cell type assignments are provided. scRNA-seq, single-cell RNA-sequencing; UMI,

unique molecular identifier.
(CSV)

**S32 Table. Mouse LPS challenge circulating immune scRNA-seq clustering data.** For each mouse LPS challenge circulating immune cell in our scRNA-seq data, the corresponding sample, number of captured UMIs, number of captured genes, first-iteration and convergence cluster and cell type assignments are provided. LPS, lipopolysaccharide; scRNA-seq, single-cell RNA-sequencing; UMI, unique molecular identifier.
(CSV)

**S33 Table. NM-R saline control circulating immune scRNA-seq clustering data.** For each NM-R saline control circulating immune cell in our scRNA-seq data, the corresponding sample, number of captured UMIs, number of captured genes, first-iteration and convergence cluster and cell type assignments are provided. NM-R, naked mole-rat; scRNA-seq, single-cell RNA-sequencing; UMI, unique molecular identifier.
(CSV)

**S34 Table. NM-R LPS challenge circulating immune scRNA-seq clustering data.** For each NM-R LPS challenge circulating immune cell in our scRNA-seq data, the corresponding sample, number of captured UMIs, number of captured genes, first-iteration and convergence cluster and cell type assignments are provided. LPS, lipopolysaccharide; NM-R, naked mole-rat; scRNA-seq, single-cell RNA-sequencing; UMI, unique molecular identifier.
(CSV)

**S35 Table. Mouse saline control circulating immune scRNA-seq cluster marker genes.** For each marker gene in each of the converged clusters of the mouse saline control circulating immune scRNA-seq data, provided are its Ensembl gene ID, gene symbol, gene description, gene biotype, and converged cluster and cell type. scRNA-seq, single-cell RNA-sequencing.
(CSV)

**S36 Table. Mouse LPS challenge circulating immune scRNA-seq cluster marker genes.** For each marker gene in each of the converged clusters of the mouse LPS challenge circulating immune scRNA-seq data, provided are its Ensembl gene ID, gene symbol, gene description, gene biotype, and converged cluster and cell type. LPS, lipopolysaccharide; scRNA-seq, single-cell RNA-sequencing.
(CSV)

**S37 Table. NM-R saline control circulating immune scRNA-seq cluster marker genes.** For each marker gene in each of the converged clusters of the NM-R saline control circulating immune scRNA-seq data, provided are its RefSeq gene ID, gene symbol, and converged cluster and cell type. NM-R, naked mole-rat; scRNA-seq, single-cell RNA-sequencing.
(CSV)

**S38 Table. NM-R LPS challenge circulating immune scRNA-seq cluster marker genes.** For each marker gene in each of the converged clusters of the NM-R LPS challenge circulating immune scRNA-seq data, provided are its RefSeq gene ID, gene symbol, and converged cluster and cell type. LPS, lipopolysaccharide; NM-R, naked mole-rat; scRNA-seq, single-cell RNA-sequencing.
(CSV)

**S39 Table. Mouse LPS challenge versus saline control circulating immune scRNA-seq expression change GSEA.** For each cluster, for each enriched gene set, provided are its category description, *p*-value, adjusted *p*-value (*q*-value), test statistic, enrichment direction (up or

down), and the Entrez IDs and symbols of the leading-edge genes. GSEA, gene set enrichment analysis; LPS, lipopolysaccharide; scRNA-seq, single-cell RNA-sequencing.
(CSV)

**S40 Table. Mouse LPS challenge versus saline control circulating immune scRNA-seq expression induction GSEA.** For each cluster, for each enriched gene set, provided are its category description, *p*-value, adjusted *p*-value (*q*-value), test statistic, enrichment direction (on or off), and the Entrez IDs and symbols of the leading-edge genes. GSEA, gene set enrichment analysis; LPS, lipopolysaccharide; scRNA-seq, single-cell RNA-sequencing.
(CSV)

**S41 Table. NM-R LPS challenge versus saline control circulating immune scRNA-seq expression induction GSEA.** For each cluster, for each enriched gene set, provided are its category description, *p*-value, adjusted *p*-value (*q*-value), test statistic, enrichment direction (on or off), and the Entrez IDs and symbols of the leading-edge genes. GSEA, gene set enrichment analysis; LPS, lipopolysaccharide; NM-R, naked mole-rat; scRNA-seq, single-cell RNA-sequencing.
(CSV)

**S42 Table. NM-R LPS challenge *Ltf*-high circulating neutrophils versus circulating neutrophils scRNA-seq expression change GSEA.** For each cluster, for each enriched gene set, provided are its category description, *p*-value, adjusted *p*-value (*q*-value), test statistic, enrichment direction (on or off), and the Entrez IDs and symbols of the leading-edge genes. GSEA, gene set enrichment analysis; LPS, lipopolysaccharide; NM-R, naked mole-rat; scRNA-seq, single-cell RNA-sequencing.
(CSV)

**S43 Table. scRNA-seq data summary statistics.** For each scRNA-seq sample provided are the species, sex, tissue, condition, biological replicate, technical replicate, number of reads, percentage of mapped reads, percentage of confidently mapped reads, percentage of confidently mapped intergenic reads, percentage of confidently mapped intronic reads, percentage of confidently mapped exonic reads, percentage of confidently mapped transcriptomic reads, percentage of antisense to gene mapped reads, number of filtered cell-less barcodes, number of multiplet filtered barcodes, number of retained barcodes, number of filtered genes, number of retained genes across all cells, and number of variable genes across the replicates. scRNA-seq, single-cell RNA-sequencing.
(CSV)

**S44 Table. scRNA-seq clusters annotation.** For each converged cluster in each of the scRNA-seq datasets provided are the Ensembl or RefSeq genes IDs (for mouse and NM-R genes, respectively), ortholog Ensembl gene ID (only for NM-R genes), gene symbol, ortholog gene symbol (only for NM-R genes), converged cluster, converged cell type, ImmGen cell type, ImmGen TPM, species, dataset, and established cell type the gene is associated with. NM-R, naked mole-rat; scRNA-seq, single-cell RNA-sequencing; TPM, transcripts per million.
(CSV)

**S45 Table. Differential number of expressed genes across NM-R spleen cell clusters.** For each differential expressed-gene count contrast (in each converged cluster), provided are the converged cluster cell type, effect size, standard error of effect size, *p*-value, and adjusted *p*-value (*q*-value). NM-R, naked mole-rat.
(CSV)

**S46 Table. Differential number of expressed genes across NM-R circulating immune cell clusters.** For each differential expressed-gene count contrast (in each converged cluster), provided are the converged cluster cell type, effect size, standard error of effect size, *p*-value, and adjusted *p*-value (*q*-value). NM-R, naked mole-rat.
(CSV)

**S47 Table. Genomes used for constructing the phyletic pattern.** For each species provided are the species name, phylogenetic group, phylogenetic group type, species scientific name, genome assembly, whether Ensembl was the source, annotation release version, name of GTF file, name of genome sequence fasta file, name of protein sequences fasta file, name of transcript sequences fasta file, name of cDNA sequences fasta file, name of ncRNA sequence fasta file, genome sequence fasta file FTP URL, transcript sequences fasta file FTP URL, protein sequences fasta file FTP URL, cDNA sequences fasta file FTP URL, ncRNA sequences fasta file FTP URL, GTF FTP URL, taxon Entrez ID, assembly level, genome representation level, genome sequence release date, Entrez assembly ID, sequencing technology, genome assembler used, and genome coverage. ncRNA, noncoding RNA.
(CSV)

**S48 Table. NM-R mouse one-to-one gene orthologs.** For each one-to-one NM-R mouse orthologous gene, provided are the NM-R and mouse RefSeq and Ensembl gene IDs, respectively, and their gene symbols. NM-R, naked mole-rat.
(CSV)

**S49 Table. NM-R and mouse LPS challenge and saline control matched spleen scRNA-seq clusters.** For each species in each condition, provided are the original first-iteration cluster and corresponding cell type, and the cluster and cell-type used when the datasets were combined for differential expression analysis, separately for each species. LPS, lipopolysaccharide; NM-R, naked mole-rat; scRNA-seq, single-cell RNA-sequencing.
(CSV)

**S50 Table. NM-R and mouse LPS challenge and saline control matched circulating immune scRNA-seq clusters.** For each species in each condition provided are the original first-iteration cluster and corresponding cell type, and the cluster and cell type used when the datasets were combined for differential expression analysis, separately for each species. LPS, lipopolysaccharide; NM-R, naked mole-rat; scRNA-seq, single-cell RNA-sequencing.
(CSV)

**S51 Table. NM-R and mouse matched LPS challenge and saline control matched spleen scRNA-seq clusters.** For each species in each condition provided are the original first-iteration cluster and corresponding cell type, and the cluster and cell type used when the datasets were combined for differential expression analysis, for the species combined. LPS, lipopolysaccharide; NM-R, naked mole-rat; scRNA-seq, single-cell RNA-sequencing.
(CSV)

**S52 Table. NM-R and mouse LPS challenge and saline control matched circulating immune scRNA-seq clusters.** For each species in each condition provided are the original first-iteration cluster and corresponding cell type, and the cluster and cell type used when the datasets were combined for differential expression analysis, for the species combined. LPS, lipopolysaccharide; NM-R, naked mole-rat; scRNA-seq, single-cell RNA-sequencing.
(CSV)

## Acknowledgments

We would like to thank Tal Pupko for his advice regarding the phylogenetic analysis, the members of the computing team at Calico for their useful advice in the analysis of the scRNA-seq data, and the members of the senior staff at Calico for their critique of the manuscript.

## Author Contributions

**Conceptualization:** Hugo G. Hilton, Nimrod D. Rubinstein, Rochelle Buffenstein.

**Formal analysis:** Nimrod D. Rubinstein, Nicole L. Fong.

**Investigation:** Hugo G. Hilton, Nimrod D. Rubinstein, Peter Janki, Kevin M. Wright, Megan Smith, David Finkle, Denise M. Imai, Vladimir Jojic, Rochelle Buffenstein.

**Methodology:** Hugo G. Hilton, Nimrod D. Rubinstein, Nicholas Bernstein, Rochelle Buffenstein.

**Project administration:** Hugo G. Hilton, Rochelle Buffenstein.

**Resources:** Andrea T. Ireland, Megan Smith, David Finkle, Baby Martin-McNulty, Margaret Roy, Rochelle Buffenstein.

**Supervision:** Rochelle Buffenstein.

**Validation:** Hugo G. Hilton, Nimrod D. Rubinstein.

**Visualization:** Hugo G. Hilton, Nimrod D. Rubinstein.

**Writing – original draft:** Hugo G. Hilton, Nimrod D. Rubinstein, Rochelle Buffenstein.

**Writing – review & editing:** Hugo G. Hilton, Nimrod D. Rubinstein, Rochelle Buffenstein.

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
