## [Editor Report · Decision Letter 0]

27 Aug 2019

Dear Dr Buffenstein, 

Thank you for submitting your manuscript entitled "Single-cell transcriptomics of the naked mole-rat reveals unexpected features of mammalian immunity" for consideration as a Short Reports by PLOS Biology.

Your manuscript has now been evaluated by the PLOS Biology editorial staff as well as by an academic editor with relevant expertise and I am writing to let you know that we would like to send your submission out for external peer review.

*Please be aware that, due to the voluntary nature of our reviewers and academic editors, manuscripts may be subject to delays during the holiday season. Thank you for your patience.*

Please re-submit your manuscript within two working days, i.e. by Aug 29 2019 11:59PM.

Kind regards,

Di Jiang, PhD

Associate Editor

PLOS Biology

---

## [Decision Letter · Decision Letter 1]

2 Oct 2019

Dear Dr Rubinstein,

Thank you very much for submitting your manuscript "Single-cell transcriptomics of the naked mole-rat reveals unexpected features of mammalian immunity" for consideration as a Short Reports by PLOS Biology. Your paper was evaluated by the PLOS Biology editors as well as by an Academic Editor with relevant expertise and by four independent reviewers. 

Based on the reviews, we will probably accept this manuscript for publication, assuming that you will modify the manuscript to address all the concerns raised by the reviewers. You are encouraged to provide additional data if they are already available to answer several questions from reviewer 3, but further experimentation won't be required for the acceptance of this paper. 

We expect to receive your revised manuscript within two weeks. Your revisions should address the specific points made by each reviewer. In addition to the remaining revisions and before we will be able to formally accept your manuscript and consider it "in press", we also need to ensure that your article conforms to our guidelines; several of which are described below under DATA POLICY and are marked with "***IMPORTANT: ". A member of our team will be in touch shortly with a set of requests. As we can't proceed until these requirements are met, your swift response will help prevent delays to publication.

Please note that you may have the opportunity to make the peer review history publicly available. The record will include editor decision letters (with reviews) and your responses to reviewer comments. If eligible, we will contact you to opt in or out.

Sincerely,

Di Jiang, PhD

Associate Editor

PLOS Biology

DATA POLICY:

***IMPORTANT: Regardless of the method selected, please ensure that you provide the individual numerical values that underlie the summary data displayed in the following figure panels: 3GH, S1BCDGIJKN, S2D, S3B, S4BCD, S7A, S8EF, S9E, S10IN, as they are essential for readers to assess your analysis and to reproduce it. ***IMPORTANT: Please also ensure that figure legends in your manuscript include information on where the underlying data can be found. You can write in the figure legends: "Underlying data can be found in S1 Data.".

***IMPORTANT: Please provide a reviewer token for GSE132642 deposited in GEO.

Reviewer remarks:

Reviewer #1 (Ewan St. John Smith, signed review): The study from Hilton et al., is the first to make a comprehensive study of the immune system of the naked mole-rat. Considering the unusual biology of the naked mole-rat, e.g. extreme longevity and general resistance to the processes associated with ageing (as demonstrated through its resistance to cancer and neurodegeneration that are both associated with ageing) it is extremely important to understand more about naked mole-rat immune system, a system that generally deteriorates with ageing.

The authors use unbiased single cell RNA-sequencing of the spleens from mice and naked mole-rats to determine the comparative constituents of the immune systems of these two species. Data acquired from analysis of mouse splenic immune cells are consistent with what others have shown (e.g. a high proportion of lymphoid lineage B and T cells and low proportion of myeloid cells), whereas splenic immune cells of the naked mole-rat form a strikingly different composition, i.e. more myeloid than lymphoid lineage cells (including two subgroups of neutrophils) and an apparent absence of natural killer cells.

Details on the absence of natural killer cells from the spleen are provided and further single cell RNA-sequencing is conducted on circulating immune cells, which, along with efforts to simulate the possible poor annotation of natural killer cell marker genes in the naked mole-rat genome, confirms that naked mole-rats really do appear to lack a natural killer cell population. This finding correlates with a lack of expansion of genes like Ly49 that are associated with natural killer cell function. A broader comparison of 45 mammalian species places the naked mole-rat data in a wider context, demonstrating that the closely related guinea pig has retained control mechanisms for natural killer cells.

The authors next address the comparative gene expression of immune cells following lipopolysaccharide (LPS) injection in mice and naked mole-rats, both species activating similar pathways. However, interestingly, among the two neutrophil subgroups in the naked mole-rat, the naked mole-rat specific high lactotransferrin group increase in number and at the same time there was a convergence of the two groups, a result suggesting that although these two neutrophil subgroups exist with regard to gene transcription, they are in fact likely playing similar roles.

Overall, this is a really exciting paper that documents for the first time the immune system of the naked mole-rat, demonstrating key differences compared to other mammals. Importantly, this work is of a wide general interest and will provoke make future avenues of research e.g. how does the immune system of the naked mole-rat deal with a variety of challenges? This manuscript clearly establishes a platform for all future work to build upon.

Minor comments that should be addressed in a revised version:

- the authors give the age and sex of animals used, they should also state throughout as to whether or not the mice used were nulliparous, and whether or not the naked mole-rats were breeding or subordinate animals? This is stated for naked mole-rats used for blood smears on line 6 of the Materials and Methods, but not, as far as I can tell, for other parts.

- naked mole-rats are naturally quite an inbred species, a situation likely pushed further through laboratory housing due to constraints of mixing and matching a species that is, as the authors note, both xenophobic and eusocial. Although such laboratory breeding would likely not explain the striking findings of this paper (e.g. absence of natural killer cells, two neutrophil subgroups and distinct LPS challenge response), it could perhaps influence how well the sequencing data obtained reflect the wild naked mole-rat immune cell transcriptome. Obviously, the same criticism could be aimed at any study using inbred mouse strains, but it would perhaps be helpful to know if animals throughout the study came from the same colony, or different colonies.

- what evidence do the authors have that markers used for different immune cell subgroups in the mouse are appropriate for use in the naked mole-rat, i.e. for the data in Figure 1A/B, Cxcr2 is used to identify neutrophils – how robust a marker is this in different species? I do not expect the authors to validate every marker, but a general discussion on this point would, I think, be helpful, especially for the more general reader.

Reviewer #2 (Pontarotti Pierre, signed review): Review of the article Single-cell transcriptomics of the naked mole-rat reveals unexpected features of mammalian immunity" by Dr. Nimrod Daniel Rubinstein for PLOS Biology.

In summary the authors show that the naked -mole rat lost the NK cells and corresponding gene during the evolution of the Heterocephalus clade . This is an important finding that can help in the understanding (give more insight) of life longevity and cancer resistance mechanisms. In this regard the choice of the naked mole-rat model is appropriate.

My field is comparative and evolutionary genomic , therefore I have focused on this part of the article . 

The in silico analyis (including comparative genomic ) is well described and robust and supports the authors conclusions .

Reviewer #3: The manuscript entitled “Single-cell transcriptomics of the naked mole-rat reveals unexpected features of mammalian immunity” describes a single cell RNA sequencing analysis of splenic cells in the naked mole rat compared to conventional mice. The interest of NMR is their apparent lower susceptibility to develop tumors. The single cell RNA approach revealed that compared to mice the NMR has fewer B cells and more myeloid cells which skews the ratio lymphoid/myeloid cells that in the mouse is more than 2:1. Two additional features came out of the analysis, one was the apparent lack of NK cells and the other was the detection of a major population of LPS responsive neutrophils not seen in the mouse. This approach is of interest to immunologists and biologists in general. However, the data is solely descriptive and falls short of bringing concrete information on the function of the NMR immune system

Too much relevance is given here to establish a link between the lack of bona fide NK cells, the higher representation of myeloid cells and the low susceptibility to tumors in NMR. The major role of NK cells is a first line of defense to viral infection due to their capacity to detect cells with low or no MHCI expression modulated by many viruses. Tumor immune surveillance has also been described but other adaptive immune cells also play a role (see Smyth et al J. Exp. Med 2000 among others). In fact, recent strategies of cancer immune therapy focus on controlling check points within the adaptive immunity rather than stimulating the ILC compartment. That NMR do not have NK cells is not of major relevance to the discussion because other adaptive immune cells can also secrete cytokines, can take the protective role of NK cells and mechanisms other than immune response related might have evolved in the NMR (cell cycle arrest mechanisms, tumor suppressor genes such as P53 and many other). More relevant in this context is the higher susceptibility of NMR to viral infection. 

Because there are fewer B cells, are immunoglobulin levels present in the serum at similar levels? Can memory B and T cells be induced by immunizing with thymus dependent antigens? Are other TLR inducing similar responses? Can antibody responses be induced? Although I understand that this is a short report these would be simple approaches to better characterize the immune system. 

Also it should be clarified why there is a large population of erythroid cells found in the NMR and not in the mouse. Because RNA was obtained by total organ lysate this might indicate that in contrast to mouse the NMR erythroid cells have high amounts of RNA. Is the spleen a site of erythroid development in the NMR?

Because the data is deposited in GEO are 48 Sup Tables needed?

Reviewer #4 (Jim Kaufman, signed review): This manuscript describes single cell RNAseq studies on spleen and blood cells of the naked mole rats in comparison with an inbred mouse strain, both unstimulated and after LPS treatment (as well as a big meta-analysis of gene copy number and a cell staining experiment). I am no expert on such single cell sequencing (although there is no escape from learning about it from many publications and seminars), but there is no question in my mind that this study is worthy of publication in a general high-impact journal of biology. Naked-mole rats are fascinating creatures with many unusual and unexplained characteristics, there is an enormous amount of work done in this paper which seems very well controlled, the results are extremely interesting and novel, the speculations will excite new functional experiments, and the manuscript is very carefully and well-written, so I would predict that this excellent paper will be cited in all studies about this fascinating creature in several different scientific fields for years to come. 

I have only two concerns for the authors to consider. 

The first concern I believe that the authors should answer. They identify macrophages, red pulp macrophages and (Timeless-high) dendritic cells in the naked mole rat spleen, but only “antigen presenting cells” (APCs) in the mouse. I find this surprising, as the APCs of mice have been well-studied over many decades and so there are certainly RP and other macrophages as well several kinds of DCs in mouse spleen. I also note that the tSNE plots of the mouse cells show a lot of structure in the APC group, so it is well-possible that all these different kinds of cells are represented in that population. Perhaps I missed the key sentence in the text, or the explanation is buried in one of the many supplementary items (of which I looked at only a few), but I think that this point deserves explanation, if only a sentence.

The second concern is more general, but I believe could be dealt with in few sentences. The authors begin their manuscript by describing some of the unexpected features of the naked mole-rat, and their motivation for the study to examine the possibility that “the NM-R has evolved features of systemic immunity that contribute to these remarkable phenotypes.” By the end of the discussion, they have certainly shown that "the NM-R has evolved unusual features of systemic immunity”, but I didn’t feel that they drew conclusions from their data about the contribution "to the remarkable phenotypes". They include three other “histricomorphs” (Damaraland mole-rat, chinchilla and guinea pig) in their meta-analysis, but do not state clearly that their discoveries of cellular composition might be unique to the naked mole-rats or, alternatively, might be an evolutionary novelty for all the “histricomorphs” regardless of their social and ecological traits. Also, they correctly point out that marine carnivores (and dogs, if I remember correctly) also have low levels of NK receptor genes, without exploring the implications (for instance, no or few NK cells apparently does not automatically mean more or less cancer, etc). There is a rich vein of interesting thoughts here with many further experiments to suggest, but I assume the authors wished to avoid a messy discussion and so compressed it all into the single sentence at the end of the discussion. As far as I am concerned, it is up to the authors, but they did leave me wishing for a bit more at the very end.

---

## [Editor Report · Decision Letter 2]

7 Nov 2019

Dear Dr Rubinstein,

On behalf of my colleagues and the Academic Editor, Avinash Bhandoola, I am pleased to inform you that we will be delighted to publish your Short Reports in PLOS Biology. 

Early Version

PRESS 

Kind regards,

Sofia Vickers

Senior Publications Assistant

PLOS Biology

On behalf of, 

Di Jiang,

Associate Editor

PLOS Biology